# Using stochastic cellular automata to model and define sufficient conditions for the survival of *Enterococcus faecalis* biofilms with the pCF10 plasmid under erythromycin treatment

**Madison Shoraka**[1], **Herby Jean-Baptiste**[2], **Bettina Buttaro**[2], **Gillian Queisser**[1]*

1 Department of Mathematics, College of Science and Technology, Temple University, Philadelphia, Pennsylvania, United States of America, 2 Department of Microbiology, Immunology and Inflammation, Lewis Katz School of Medicine at Temple University, Philadelphia, Pennsylvania, United States of America

* queisser@temple.edu

**Data availability statement:** All data and associated code is available in the main text or

## Abstract

A biofilm is a community of microorganisms adhered to a surface, bound together by extracellular polymeric substances (EPS). They are ubiquitous in nature and develop on a range of surfaces including living tissue. Biofilms themselves typically do not negatively affect their host, but under certain conditions they can retain pathogenic features and cause a wide range of illnesses including persistent or chronic infections. In this study, we look at the bacterium *Enterococcus faecalis*. *E. faecalis* is a gram-positive, commensal bacterium commonly found in the human gastrointestinal tract. Generally, commensal *E. faecalis* does not negatively impact human health, but pathogenic strains have been found to acquire mobile genetic elements, including plasmids. When *E. faecalis* with the pCF10 plasmid forms a biofilm it constructs raised complex structures with variable cellular packing, including aggregates, instead of a homogeneous and less densely packed biofilm above a rigid base. This reconfiguration of the biofilm confers resistance to high levels of erythromycin. For this study, we carried out biological experiments which show that pCF10-containing *E. faecalis* biofilms undergo a rapid reconfiguration of its initial architecture, resulting in a doubling of cellular population over a single hour of antibiotic treatment. We developed a mathematical and computational model, calibrated using image processing techniques, to identify the characteristics of the biofilm's spatial architecture that allow for the rapid one-hour reconfiguration under treatment. This model involves both stochastic cellular automata and deterministic partial differential equations. The numerical simulations carried out in this study demonstrate that biofilm survival requires both the robust formation of initial complex structures and an associated extracellular DNA (eDNA) cloud. These findings highlight the fundamental role of biofilm heterogeneity, containing aggregated structures with an associated eDNA cloud, in erythromycin resistance of *E. faecalis* with the pCF10 plasmid. The identification of eDNA

the supporting information. It is also hosted on Github found at:

https://github.com/Madison-Shoraka/E_faecalis_erm_treatment.git.

**Funding:** This work was supported by the Catalytic Collaborative Funding Initiative which is an internal grant provided by Temple University's Office of the Vice President for Research. H.J.-B. was supported by a supplement NIH grant from the parent grant of Satya Kunapuli (HL155694). The funders had no role in study design, data collection and analysis, decision to publish, or preparation of the manuscript.

**Competing interests:** The authors have declared that no competing interests exist.

as a target to increase the susceptibility of the biofilm to erythromycin could ultimately improve antibiotic treatment protocols.

## Author summary

Biofilms are communities of microorganisms bound together by a compilation of organic polymers and adhered to a surface. They are ubiquitous in nature, forming on a myriad of surfaces such as the human gastrointestinal tract, industrial piping, or natural bodies of water. Normally, they are not a danger to human health, but in some cases, the microorganisms that live in biofilms can attain pathogenic traits. *E. faecalis* is an example of a microorganism that is generally harmless and found in most people's gastrointestinal tract, but when they acquire certain pathogenic traits such as antibiotic resistance genes, their biofilm formations have been implicated in human health maladies that range from urinary tract infections to endocarditis. In this study, we assessed the behavior of resistant *E. faecalis* under antibiotic treatment. Biological experiments show that pCF10-containing *E. faecalis* biofilms undergo a rapid reconfiguration of its initial architecture resulting in a doubling of cellular population over a single hour of antibiotic treatment time. Using mathematical modeling and image processing techniques we identified the necessary features of the initial biofilm architecture that are sufficient for pCF10-containing *E. faecalis* to undertake this antibiotic-resistant reconfiguration.

## 1. Introduction

A biofilm is a community of microorganisms adhered to a surface and held together by extracellular polymeric substances (EPS). They are widespread in nature and develop on a range of surfaces both artificial and natural. Biofilms themselves typically do not negatively affect their host environment. In some cases, they are beneficial, such as in wastewater treatment, where they help break down organic matter [1] or in the human gut microbiota, where they assist the host with a diverse range of tasks [2]. In other cases, biofilms can retain pathogenic features and cause a wide range of medical conditions, including approximately 60%–80% of microbial-associated human infections [3,4]. Bacterial communities are highly plastic and can attain pathogenic features, including resistance to antibacterial agents. These resistant colonies are then able to cause untreatable and often deadly infections [5–14]. In general, the frequency of finding resistant bacterial strains is on an upward trend with no end in sight.

The bacterium of interest in this work is *Entercocous faecalis*, a gram positive ubiquitous commensal strain found in the human gastrointestinal tract. Generally not a danger to its human host, *E. faecalis* can become a risk to human health when it gains mobile genetic factors, such as plasmids [15]. When in a pathogenic state, *E. faecalis* has been known to cause infectious endocarditis, septicemia, and urinary tract infections in the most vulnerable populations [15]. These infections can often lead to death, particularly when *E. faecalis* has acquired resistance to multiple antibacterial agents. In particular, pheromone-responsive plasmids have been identified as a prevalent element in clinical isolates [16]. In the presented work, the plasmid responsive to pheromones pCF10 is introduced into the commensal strain OG1RF of *E. faecalis* thereby increasing its virulence [17,18]. The pCF10 plasmid

and related pheromone signaling systems of *E. faecalis* are well studied topics at the single cell and population levels [19–25]. Although there is no known gene encoding erythromycin resistance on pCF10, in these biological experiments the plasmid is able to confer resistance to the antibiotic erythromycin. All experiments conducted in this project have been carried out using the *E. faecalis* commensal strain OG1RF or OG1RF containing pCF10 [OG1RF(pCF10)].

Under antibiotic stress, *E. faecalis* biofilms are known to alter their structural composition [26], and the strain of our interest is no different. Under treatment by erythromycin, we have observed that OG1RF(pCF10) undergo a rapid one-hour structural remodeling. When forming a biofilm, OG1RF(pCF10) constructs small complex structures with variable cellular packing. Laboratory experiments, detailed below, suggest the presence of these preformed structures, containing aggregated regions which form protective walls that enclose less densely packed areas. We show that these aggregated structures surrounded by lower density biofilm are a necessary condition for sustained biofilm growth and protection during erythromycin treatment. Further, we found that these protective structures tend to double in size over the course of treatment. We therefore pose the question whether initial structure formation alone is also a sufficient condition for biofilm survival or if other biofilm characteristics play an important role. To answer this question we developed a data-driven cellular automata model.

The ubiquitous nature of biofilms has precipitated the necessity of studying their dynamics at a mechanistic level. Mathematical modeling is one of the tools used to assess and predict a biofilm's internal behavior and how it interacts with its environment. In the past, biofilm modeling has been done at the discrete [27–33] and continuum levels [34–53]. Mathematical models have been employed to capture a variety of biofilm dynamics, including growth dynamics [30,35,36,38,43,49], quorum sensing [32,38,40], structure formation [30,34–37], molecular diffusion [33,35,42], and antimicrobial resistance [40,41]. Although continuum models have been recently favored over discrete models, there has been a resurgence of using individual-based cellular automata models to capture cellular interactions at higher resolution. This has been driven by the increased availability and resolution of experimental data. When it comes to investigating the heterogeneous organization of biofilms, existing work has employed discrete models to study the differences in metabolic activity [33], competition between two-species [31], and variation of genetic mutations [28].

In contrast to the previously mentioned studies of heterogeneity in biofilms, research presented here makes use of stochastic cellular automata to study the heterogeneity of cellular density and/or aggregation that is seen in *E. faecalis* OG1RF(pCF10) biofilms as they alter their structural composition in response to antibiotic stress [26]. The developed mathematical model is driven by experimental data including microscopy images of biofilms and time series assays of colony forming units before and post treatment. A growth model was chosen because cell numbers measured in colony forming units and imaging supports doubling of structure size. Despite extensive studies on *E. faecalis* biofilms, to these authors' knowledge, neither dispersal nor a dispersal mechanism for *E. faecalis* has been reported. To capture this growth-driven reconfiguration process, we allow nearby antibiotic concentrations to modulate local cellular density. Mathematically, the developed model is akin to the modeling concept of swarming, but rather than movement-driven swarming (which is seen with flocking birds or schooling fish), we investigate growth-driven swarming.

Our simulation results suggest that preformed complex structures are not sufficient for the survival of *E. faecalis* OG1RF(pCF10) under erythromycin treatment. We instead show that in addition to the preformed complex structures, the eDNA cloud, which is formed during growth of a biofilm, must cover them. Similar to a lid, the eDNA cloud that covers the

preformed structures appears to provide both a reaction and diffusion barrier. We show that there exists an extensive eDNA cloud above the complex structures in our experiments. This attribute is incorporated into the model through the boundary conditions which modulate the antibiotic flux into the computational domain. We determined that an adequate eDNA cloud covering and initial structure formation is both necessary and sufficient for the biofilm's survival under erythromycin treatment. Various studies have already implicated the production of eDNA by various strains of *E. faecalis* biofilm to be a critical virulence factor [54,55]. Our work further justifies the critical role the eDNA cloud plays and points to a reason for this increased virulence. Ultimately, these results could have an impact on the understanding of antibiotic resistance and possibly improve treatment strategies.

## 2. Results

To establish sufficient survival conditions of *E. faecalis* OG1RF(pCF10) under erythromycin treatment, we evaluated the conservation of protected regions over the observed rapid one hour remodeling. Protected regions are defined as less densely packed areas of the complex structure that are buffered by at least two or more higher density microstructures. These higher density microstructures are then referred to as protective structures (see Sect 4.4). In Sect 2.2, we show that protected regions are conserved during treatment using an image processing algorithm applied to microscopy stacks of treated and untreated biofilms. This conservation is then used in Sect 2.3 to derive the necessary boundary conditions for antibiotic diffusion. Subsequently, we validated our model and conclude that the initial complex structures alone are not sufficient for biofilm survival.

### 2.1. Biological experiments reveal initial structure formation is necessary for survival under antibiotic treatment

The experiment that drove our modeling process consisted of using erythromycin to treat formed OG1RF *E. faecalis* biofilms with and without the pCF10 plasmid. We imaged the biofilm's structural composition and counted the colony forming unit (CFU) before and after one hour of treatment. As seen in Fig 1 and S1 Data, consistent with previous studies, OG1RF formed a homogeneous biofilm consisting of a rigid densely packed base layer overlaid by a less-densely packed biofilm (Fig 1A). Treatment with 100 times the minimum inhibitory concentration (100X MIC) of erythromycin reduced the number of bacteria in the OG1RF biofilms by an average of 2 orders of magnitude, while pCF10-containing OG1RF [OG1RF(pCF10)] showed a doubled in CFU (Fig 1B). Complex structures directly impacted the ability of the biofilm to mount a rapid response during treatment with the bacteriostatic antibiotic erythromycin (Fig 1C–1G). This led to the conclusion that initial structure formation due to the presence of the plasmid pCF10 is necessary for the survival of *E. faecalis* OG1RF biofilm under erythromycin treatment.

Further, it should be noted that this remodeling takes place via growth, not movement, as we see a general doubling in CFU counts over treatment (see Fig 1). Less densely packed cells experience movement associated with the viscoelastic properties of the EPS matrix [56]. Thus, although they move with respect to the plate, they do not move with respect to each other. Besides the cellular death process, bacterial cells do not abandon their respective embedding in the matrix. Finally, to the knowledge of these authors, despite extensive study by multiple investigators, dispersal of *E. faecalis* biofilms has not been described. In Sect 4.1 we develop a model to gain a mechanistic understanding of this rapid reconfiguration. In a subsequent step, the model is used to determine whether these initial structures alone are sufficient for biofilm survival, or if other conditions or components are also required.

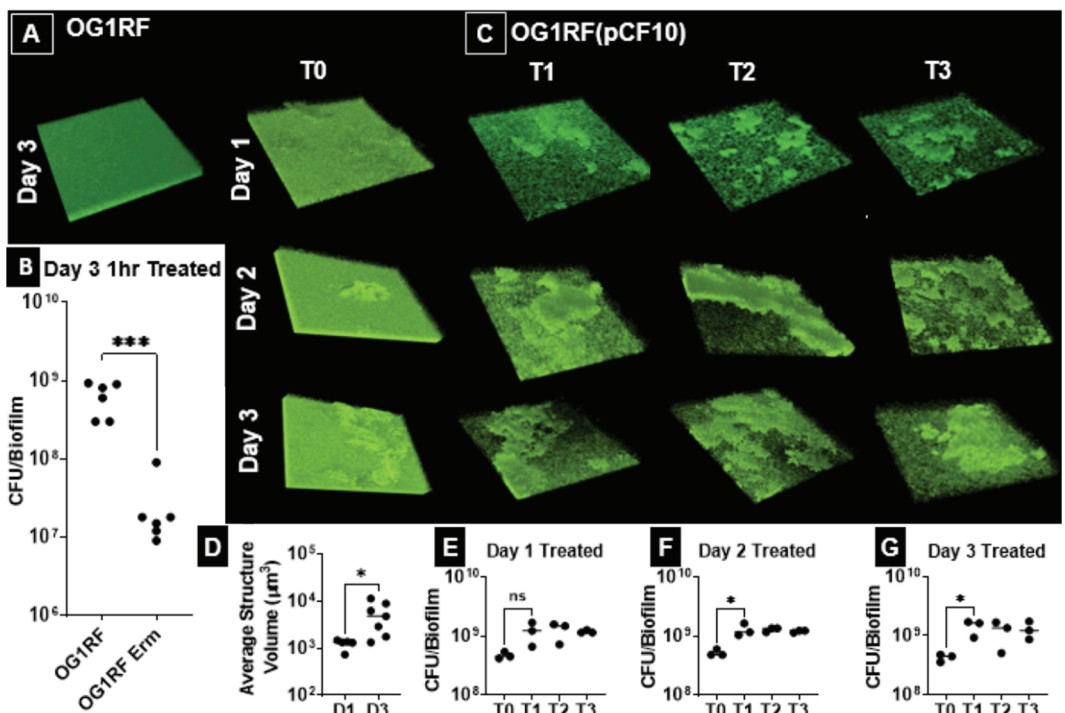

**Fig 1. Effects of the plasmid pCF10 on OG1RF biofilm rapid remodeling and growth under treatment with 100X MIC erythromycin.** Biofilms were grown in 24-well optic bottom plates or on glass coverslips in 24-well plates for with twice-daily fresh Todd-Hewitt medium exchanges for indicated times. For microscopy images, biofilms were stained with Syto9 and imaged by laser scanning confocal microscopy (LSCM). Bacterial numbers were determined by sonication followed by serial dilution and are reported as colony forming units (CFU) per biofilm. Erythromycin (100X MIC) was added to the biofilms for indicated times. **(A)** OG1RF 3-day old biofilm. **(B)** CFU of Day 3 OG1RF biofilms treated for 1 hr with TH or 100X MIC erythromycin for 1 hr. **(C)** OG1RF(pCF10) biofilms (Row 1) 1 day, (Row 2) 2 day, (Row 3) 3-day biofilms. (Column 1) 1 hr after the addition of TH only, (Column 2) 1 hour post 100X MIC erythromycin treatment, (Column 3) 2 hours post treatment, and (Column 4) 3 hours post treatment. Each image is representative of a total of 3 images for 3 biological repeats at each time point for a total of 39 biofilms. **(D)** Change in the size of structures in untreated biofilms between Day 1 and Day 3. **(E)** Change in CFU/biofilm after erythromycin treatment for Day 1, **(F)** Day 2, and **(G)** Day 3 biofilms. Each dot represents a biological repeat. Note that one star implies a p-value less than .05, and three stars implies a p-value less than .0001.

## 2.2. Image processing reveals conservation of protected regions over treatment

To establish necessary requirements for antibiotic entry into the studied domain, we constructed an image processing algorithm with the goal of measuring the percentage of slices of an image stack containing a protected region and, of those slices, the volumes of the protective structures. Prior to constructing the model in two dimensions, we assessed the structures for radial symmetry of protected regions which was well satisfied. We conducted this assessment by performing the image processing routine on x-slices and comparing them to the corresponding, intersecting, y-slices. This symmetry is discussed further in Sect 3 and going forward all analysis is carried out on the x-slices. An example of the image processing routine can be seen in Fig 2A–2E, and the workflow of this algorithm is as follows:

1. **Input:** 3D microscopy stack in .tif format (see Fig 2A and supporting information files S12-S26 Microscopy).

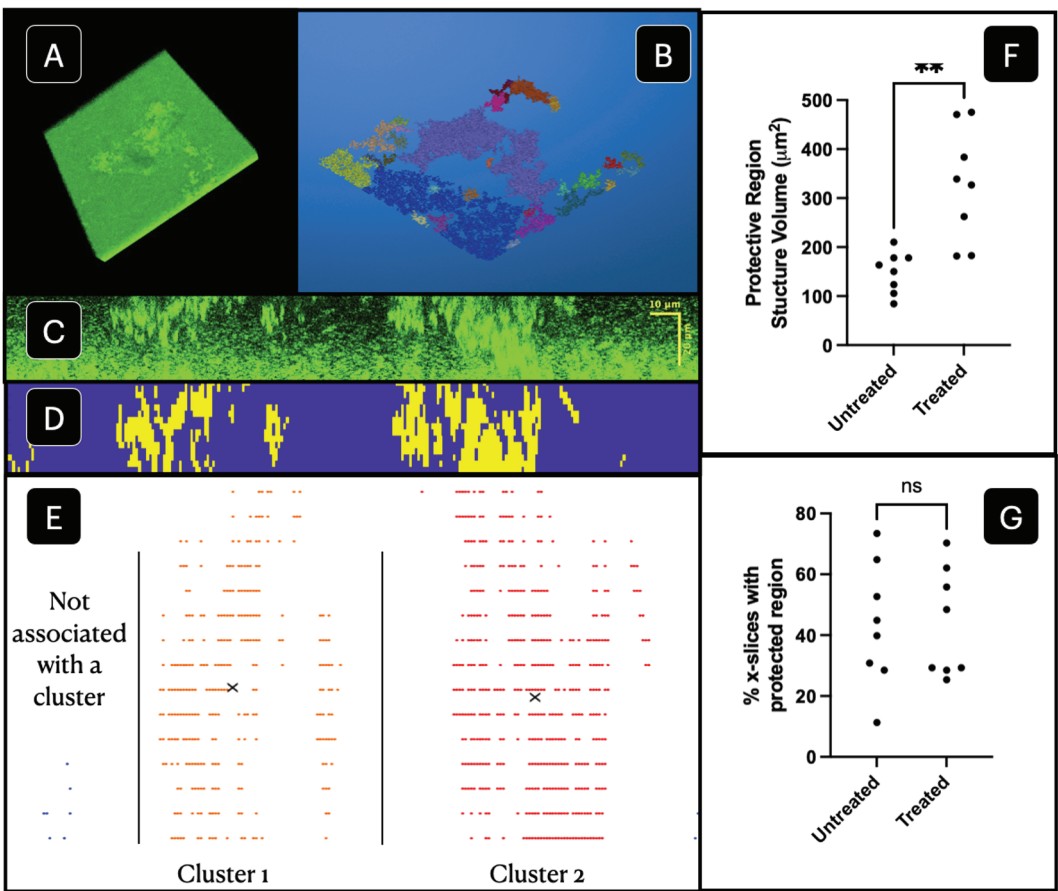

**Fig 2. Image processing steps and results. (A)** Microscopy image stack. **(B)** Post connection algorithm and removal of rigid base. **(C)** Microscopy x-slice example containing a protected region. **(D)** Corresponding x-slice in workflow. **(E)** Density-based clustering of x-slice with marked centers of mass. **(F)** Average volume of protective regions per replicate. Protective structures on average double during treatment. Double star significance indicates a p-value <0.01. **(G)** Percentage of x-slices containing a protected region per replicate, which remains unchanged during treatment. Image processing data results from 3 biological repeats with 2-3 replicates per experiment.

2. **Rigid base removal:** The rigid base layer is removed using a cutoff density threshold. When plotting the average density of z-slices, the threshold can be identified by the approximate inflection point of the curve. Values between 0.25–0.35 were found to be the range of density thresholds to remove the rigid base.

3. **Connection algorithm:** To isolate aggregated regions, a connection algorithm is run on the three-dimensional microscopy stack. Thus, only areas with high cell density, therefore strong connectivity, are identified using methods from [57] (see Fig 2B).

4. **X-slices:** Connected structures are sliced in x-direction, creating a series of two-dimensional x-slices to be evaluated (see Fig 2C and 2D).

5. **Density-based clustering:** Given an x-slice, its gray values are transformed into a point cloud on which a density-based clustering algorithm can be run [58]. This identifies regions of protection (see Fig 2E).

6. **Protective structure measure**: Number of clusters, the amount of cells within each cluster, and the distance between the clusters, is quantified. From this it can be determined whether there exists a sufficient protective region as described in Sect 4.6.1.

7. **Volume storage:** If a protective region is determined to exist for the given x-slice, the volume of each cluster (i.e. the number of cells contained in each individual cluster) is stored.
8. **Repeat:** Steps **5-7** are repeated for all x-slices of the microscopy stack.
9. **Output:** average volume of protective structures and percentage of x-slices with a sufficient protected region (see Fig 2F and 2G).

We analyzed nine microscopy images obtained from three biological repeats, each with 2–3 replicates, for both untreated and treated populations. Using our image processing algorithm, we quantified these images, yielding the results shown in Fig 2F (S2 Data) and Fig 2G (S3 Data). Notice that there is no significant change in the abundance of protected regions between treated and untreated biofilms (Fig 2G). Therefore, the antibiotic entry conditions of the model need to be chosen such that protected regions are conserved under treatment. The model can then be validated using the statistically significant data showing that the volume of protective structures roughly doubles under treatment (Fig 2F). The code for this image processing algorithm can be found in the supporting information under S1 Code.

## 2.3. Biological survival requires eDNA

We began constructing the model with the intention of determining whether the initial complex structures alone are a sufficient condition for continued growth of *E. faecalis* OG1RF(pCF10) biofilms under erythromycin treatment. In this section, we show that the answer to our question is no, initial complex structures are not sufficient. Instead, we find that initial complex structures along with the presence of an eDNA cloud are sufficient. We should first note that eDNA is present and has been observed to reside directly above the complex structures as illustrated in Fig 3A and 3B. Extracellular DNA appears as darker homogeneous regions without bright puncta concentrated above the complex structures in the 3D microscopy images and as faint staining lacking puncta in the 2D image. The eDNA also stains with Propidium Iodide (PI) routinely used to stain eDNA. A brightly PI staining subpopulation of cells associated with the eDNA was sometimes observed (white arrow), which likely represent dead cells lysing and contributing to the eDNA layer. This has previously been observed for *Pseudomonas putida* [59]. The eDNA layer in OG1RF biofilms presents itself as a homogeneous, diffuse layer. It may confer partial protection as the CFU of the OG1RF biofilm is only reduced 2 orders of magnitude in the presence of 100X erythromycin (see Fig 3B). Taken together this suggests eDNA is a possible modulator of antibiotic entry because

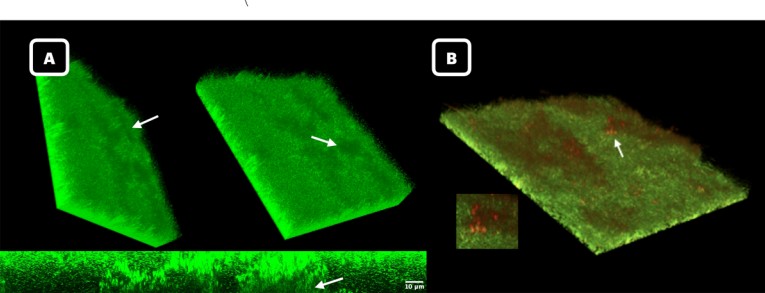

**Fig 3. Illustration of eDNA above complex structures. A.** Microscopy image capturing bottom layer of eDNA cloud (arrows) sitting atop complex structure. **B.** Microscopy image stained with propidium iodide, confirming the presence of eDNA. Microscopy images are of two independent untreated, 3-day biofilm.

it clearly resides above the complex structures, and has recently been shown to contain tRNA when derived from lysed cells (which would provide a direct binding target for erythromycin [60–63]).

The sufficiency result stems from calibrating how antibiotic molecules enter the computational domain, the only model parameter not determined in Sect 4.4. Mathematically the formulas defining this entry are encoded as boundary conditions (for details see Sect 4.1.4). By identifying the antibiotic boundary conditions that reproduce experimental results, we tested whether biofilm survival depended on characteristics other than initial structure formation. Specifically, we were interested in the influence of eDNA (green staining) and EPS cloud (not stained) above the domain (Fig 3A) on overall survival.

The computational domain, $\Omega$, is a 2D rectangle that represents an x-slice of a 3D microscopy stack. The boundaries of $\Omega$ are denoted $\partial\Omega_i$, $i = 1, 2, 3$ where $\partial\Omega_1$ is both the left and right sides of the domain, $\partial\Omega_2$ is the top of the domain (where the eDNA cloud would be located), and $\partial\Omega_3$ is the glass bottom of the optic well. An illustration of this domain can be seen in Fig 4.

We began by computing and validating antibiotic entry, i.e. boundary, conditions that, on average, conserve protected regions. The following conditions were implemented and results were compared for slices with protective regions in Fig 5:

1. **eDNA is neither a reaction nor diffusion barrier**: Antibiotics entering the domain are not influenced by a reaction or diffusion barrier, which constitutes a *bulk boundary condition*. Mathematically, antibiotic concentration is held constant on $\partial\Omega_2$. Thus, we define $a_1$ as the 100X MIC concentration and $\vec{a}(\vec{r}, t) = a_1 \ \forall t, \forall \vec{r} \in \partial\Omega_2$.

2. **eDNA is a diffusion barrier, but not a reaction barrier**: If eDNA acts as a strong diffusion barrier, antibiotics cannot diffuse from above during the hour-long treatment. However, without eDNA being a reaction barrier, erythromycin (100X MIC) can still enter the domain laterally via $\partial\Omega_1$. Mathematically, we define $a_2$ as the number of antibiotic molecules that represent 100X MIC over the course of a one hour treatment window. Then $\forall \vec{r} \in \partial\Omega_1$ and $\forall t \in [0, 1]$, $\nabla\vec{a}(\vec{r}, t) \cdot \vec{n} = a_2$.

3. **eDNA is both a diffusion barrier and a reaction barrier**: If the eDNA is both a strong reaction and diffusion barrier we assume no antibiotic flux across $\partial\Omega_2$ and a reduced flux laterally across $\partial\Omega_1$. Mathematically, this results in $a_r \in \mathbb{R}_{\geq 0}$ where $\nabla\vec{a}(\vec{r}, t) \cdot \vec{n} = a_r$ $\forall t, \forall \vec{r} \in \partial\Omega_1$. We chose $a_r$ to be the maximum number of antibiotic molecules that still preserves the protected regions. For the computational experiment in Fig 5 we found the reduction to be approximately 20%, meaning $a_r = \frac{a_2}{5}$.

Fig 5A shows three microscopy slices (supporting information files S1, S2, and S3 Microscopy), which contained protected regions, chosen from independent biological repeats.

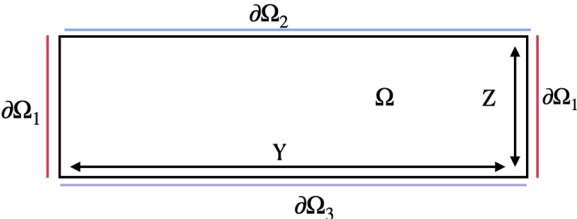

**Fig 4. Illustration of the computational domain.** Interior of the computational domain $\Omega$, is bounded by $\partial\Omega_1$ on the left and right, by $\partial\Omega_2$ on the top, and by $\partial\Omega_3$ on the bottom.

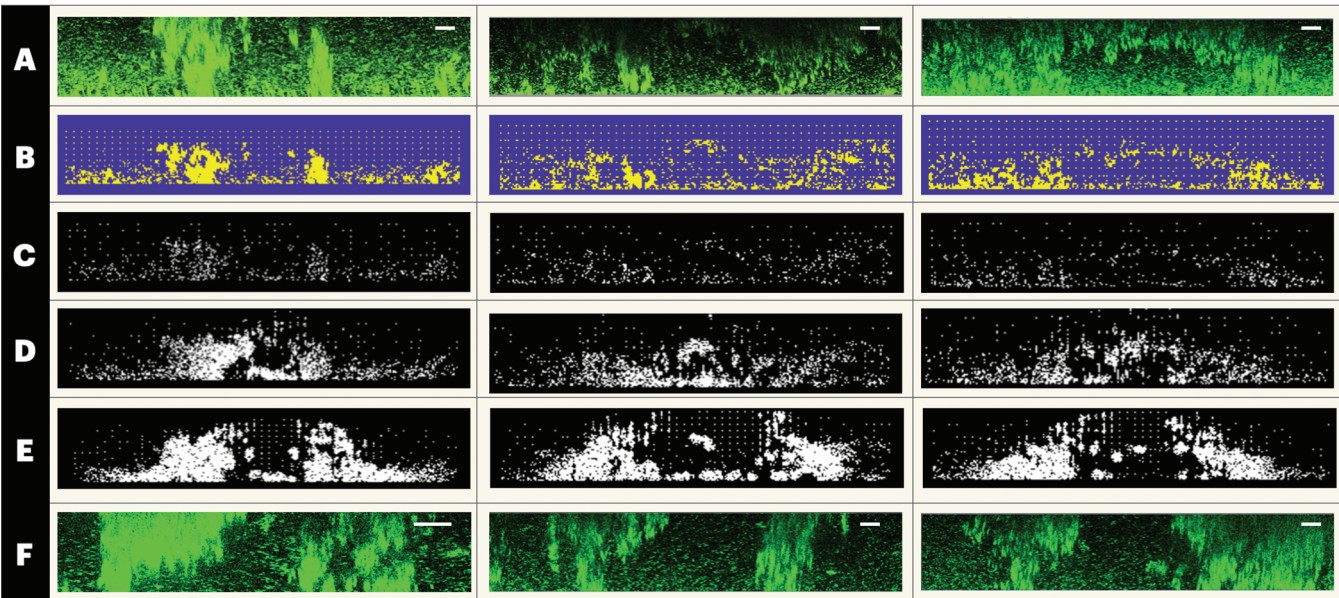

**Fig 5. Simulation results for different antibiotic entry conditions. (A)** Untreated microscopy x-slices selected using the quantification algorithm (Sect 4.6) to identify protected regions. **(B)** Initial conditions using (A) as source data. Note that in order for the cells to respond to antibiotic influx an extended vertical domain is required, since the stress distance ($\mu_d$) is 15 $\mu m$. Therefore, we added an extra 2 layers of uniformly distributed cells with a non-stressed carrying capacity above the complex structures as is seen in microscopy images. **(C)** Simulation results for scenario 1 (no reaction or diffusion barrier). **(D)** Simulation results for scenario 2 (diffusion barrier but no reaction barrier). **(E)** Simulation results for scenario 3 (diffusion and reaction barrier. **(F)** Post-treatment biofilms exposed to erythromycin for one hour. Note that the initial- and post-treatment biofilms are not the same samples. Scale bar has length 10 $\mu m$. For videos of simulations see supporting information S1 and S2 Video.

These microscopy stacks were used as the initial conditions shown in Fig 5B. The code used to generate these initial conditions can be found in the supporting information under S3 Code. Fig 5C and 5D show the simulation results when implementing the three different antibiotic entry scenarios. The simulation code can be found in supporting information under S2 Code.

In Fig 5C, we observed that simulations using scenario 1, i.e. the bulk boundary condition, for antibiotic molecules are unable to conserve protected structures, even in slices with larger than average protective structures. Fig 5D shows similar behavior using scenario 2, i.e. eDNA acts as a diffusion, but not reaction, barrier. Although a weak formation of a protective environment can be seen, these would not be classified by the image processing algorithm as protective/protected regions and ultimately would not survive treatment. Finally, in Fig 5E, we conclude that the simulations using scenario 3, i.e. eDNA is both a reaction and diffusion barrier, are able to capture the conservation of protected regions. For comparison, Fig 5F contains x-slices of three treated biofilms whose compositions look very similar to those that evolved in Fig 5E.

In conclusion, our simulations demonstrated that the presence of eDNA above rigid structures as a diffusion and reaction barrier to antibiotic molecules is a necessary condition for continued formation and growth of protected and protective micro-environments. Interestingly, the proposed eDNA reaction-diffusion barrier is biologically supported by recent observations. Studies have found that eRNA is contained in the eDNA cloud including 23S RNA [60–63], a known binding target of erythromycin [64].

To justify our simulation results, we validated the model quantitatively (see Fig 6(i)) by comparing simulation results to the image processing results found in Fig 2. In order to compare the initial conditions with their resulting simulations, we redesigned the previous three-dimensional image processing algorithm to function with two space dimensions. It should be emphasized that, since we are only working with a two-dimensional slice, we do not have access to any global parameters of the three-dimensional configurations. Without the global density parameters used in the removal of the rigid base layer, the volumes of the protective structures will typically be larger. Eight slices were randomly chosen, each from different untreated biofilms, such that they contained a protected region according to the previously defined image processing algorithm (see Fig 6(ii)A and supporting information S4-S11 Microscopy). The slices' corresponding computational estimations are shown in Fig 6(ii)B, and the result of the simulations are presented in Fig 6(ii)C. The code producing these simulations can be found in the supporting information under S4 Code.

As shown in Fig 6(i) (S4 Data), the volumetric changes of protective structures correctly capture the experimentally observed doubling when comparing untreated and treated biofilms. These results serve as validation of the model predictions that classify the eDNA cloud as an antibiotic diffusion and reaction barrier.

Lastly, we wanted to confirm whether inadequate slice selection would also produce expected results. Recall that inadequate slices are defined as biofilm x-slices that do not contain a protected region. Given that we expect adequate x-slices to be conserved under treatment, we would also expect the same conservation to hold for inadequate x-slices. We took three inadequate x-slices of varying initial total biomass (see Fig 7A) and used these to compute the corresponding computational initial conditions (see Fig 7B). The simulation results of these slices can be seen in Fig 7C in which we observe this conservation. Therefore, the model can be applied to any biofilm x-slice, whether inadequate or adequate, so long as an eDNA cloud cover is present.

In conclusion, we have shown that, although initial complex structure formation is necessary for biofilm survival and continued growth, it is not sufficient. Further, our numerical simulations and subsequent image processing results implicate the eDNA cloud above

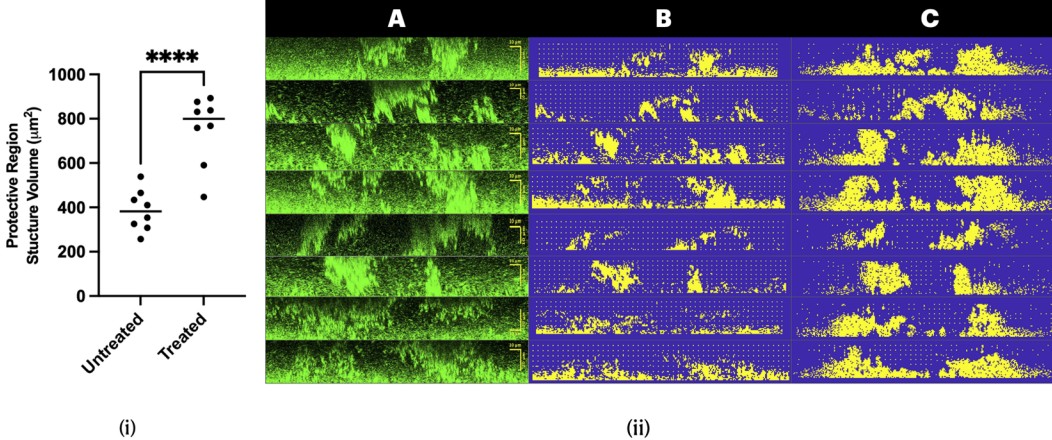

**Fig 6. Validating the computational model. (i)** Volumes of simulated protective structures confirming the doubling of protective regions. **(ii)A** Microscopy image slices used for image processing. Images were sourced from untreated 3-day biofilms grown according to Sect 4.2.**(ii)B** Initial conditions for antibiotic treatment simulations. **(ii)C** Simulation results showing rigid structure growth under stress.

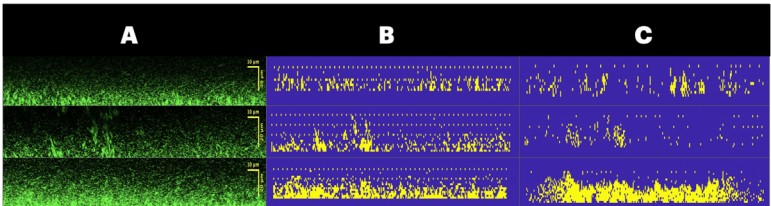

**Fig 7. Simulation results for cases where preformed structures are absent. (A)** Examples of slices without preformed structures that would be determined to be inadequate following the process described in Sect 2.2. **(B)** corresponding initial conditions for simulation. **(C)** simulation results.

complex structures as the likely addition to the survivability criterion. Taken together, our results strongly suggest that initial complex structures with an eDNA cloud functioning as an antibiotic diffusion and reaction barrier are a sufficient condition for biofilm survival and could bear relevance for further studying antibiotic resistance phenomena.

## 3. Discussion

By constructing, calibrating, and validating a stochastic cellular automata model we have captured the biological results of the experimentally observed rapid one-hour reconfiguration undertaken by *E. faecalis* biofilms with the pCF10 plasmid OG1RF(pCF10) under erythromycin treatment. We have shown that preformed structures alone are not sufficient for survival. Further, the resulting antibiotic entry conditions identified by mathematical modeling imply that to create a sufficient condition for survival, a biofilm's eDNA cloud must function as both a reaction and diffusion barrier. This requirement is consistent with the known presence of eRNA in eDNA layers, given that erythromycin inhibits protein synthesis by binding to the 50S ribosomal subunit, where the drug interacts with the unpaired bases 2058A and 2059A in the peptidyl transferase loop of 23S ribosomal RNA [64]. The presence of ribosomal RNA has been reported as a component of eRNA found mixed with eDNA surrounding biofilms [60–63]. Release of eDNA from *E. faecalis* occurs by fratricide, which would release ribosomal RNA content as well as DNA. These biofilms visually have a remarkable amount of eDNA above rigid structures, which supports the hypothesis of an eDNA cloud functioning as a diffusive and reactive barrier to antibiotics. The fact that the biofilms quickly remodel within one hour in addition to the presence of a large eDNA cloud, particularly associated with the complex structures, would support another timely hypothesis that the biofilm is in balanced steady state growth in the pretreatment phase.

The developed model bridges the temporal evolution between experimentally accessible biofilms in pre- and post-treatment state. It is not possible to carry out pre- and post-treatment imaging on the same biofilm as staining and imaging damages the biofilms. Further, we found fluorescent reporters mCherry and GFP slow bacterial growth by a factor of 30% and reduces initial formation of rigid structures by 24 hours which prohibits their use in this case, as the rapid reconfiguration is a growth dominated process. The developed image processing tool and computational model thus enabled us to predict and validate the conditions under which biofilms can survive antibiotic treatment, by continued growth and reorganization into complex, protective structures, under the protection of an eDNA cloud.

It is important to note that although eDNA is present in OG1RF *E. faecalis* without pCF10, it is not fully protective (Fig 7). The eDNA cloud acts as a stop gate and as a result the local

density of the biofilm increases prior to antibiotic penetration. Without the pCF10 plasmid, there is no aggregation response. So, any advantage provided by an extended initial antibiotic exposure period would not allow for the same reconfiguration response and, therefore, would not provide the wild-type bacteria with the same protections.

Our model and subsequent validation suggests the presence of conserved rules governing biofilm formation, organization, and growth. Since aggregated structures in biofilms can be found in multiple settings, including other pathogenic bacteria, microbiota, oral biofilms, and environmental biofilms, their structural characteristics that provide protection may be conserved throughout these systems. In mixed species biofilms, species producing aggregated structures could be able to protect other species that grow as less densely packed biofilms, helping to conserve the microbiota during antibiotic treatment. Different stressors likely interact with these structures in different ways (e.g., erythromycin binding to the eDNA layer). In the future it is feasible to employ the techniques developed here – combining biological imaging, image processing, and mathematical modeling – to show that the biofilm spatial organization described in this paper are found in a wide array of other conditions. Ideally, this would provide insight into how spatial organization of a biofilm impacts the community's tolerance to stress.

Following this result, we will next take experimental steps towards investigating how eDNA interacts with erythromycin and is modulated by the biofilm. This would include biological experiments confirming the presence of ribosomal RNA in the eDNA cloud alongside experiments to confirm the role of eDNA in protection under erythromycin treatment. Erythromycin is not known to bind polysaccharides such as rhamnose, glucans, galactose, and N-acetylglucosamine in the ECM of *E. faecalis* [65]. If eDNA is found to not contribute to the overall resistance, although unlikely, it is possible that changes in the charge or packing of the EPS matrix due to aggregation accounts for the effect. Mathematically, we are currently developing a continuum-level description of the cellular automata model to run simulations on larger, more complex computational domains. Cellular automata computationally scale poorly with increasing domain size, therefore a continuum model will be necessary. Once this continuum approximation is validated, a further application of this model would be to simulate this rapid reconfiguration of biofilms under flow. Flow cell microscopy images are shown to create much larger protective regions behind massive rigid structures. Studying biofilm behavior under flow, mimicking specific human-body environments in such a computationally assisted approach, could provide advanced methodology to studying clinically relevant states.

## 4. Models and methods

### 4.1. Model development

The overarching motivation for constructing this mathematical model is to capture the mechanisms pertaining to the experimentally observed rapid one hour remodeling under erythromycin treatment of *E. faecalis's* OG1RF(pCF10). To capture the spatial heterogeneity of the biofilm, the model was developed in two space dimensions, and therefore, all units reflect this attribute. The model tracks three variables: bacterial cells, stress, and antibiotic molecules. Bacterial cells and stress are modeled by their presence (value of 1) or absence (value of 0) at lattice nodes in the computational domain. Biologically, the induction or switching from 0 to 1 of stress is the process of aggregation substance induction undertaken by the cell. Aggregation substance is a stable cell wall protein that binds lipoteichoic acid on adjacent cells and is an irreversible, discrete, and stable process [66]. A cell expressing aggregation substance is sticky and will not fully separate from any newly divided cell it produces thereby reducing

the distance used for cellular division. The stress variable dictates the cell-to-cell distance that a given bacterial cell will require for division. Stressed cells will require less room to divide. Lastly, antibiotic molecules have the ability to walk randomly along the lattice of the computational domain, entering and interacting with bacterial cells and modulating the population of stressed cells. To formalize these interactions, we utilized the framework of stochastic cellular automata to track changes in bacterial cell placement and stress location combined with a partial differential equation (PDE) for antibiotic diffusion.

Previous attempts to directly construct a PDE model for bacterial growth with an associated stress response were futile as the cell-to-cell interactions were not well enough understood. Thus, we pivoted to using the framework of cellular automata which has given us unparalleled abilities to parse and validate cell-to-cell interactions that lead to the globally observed system behavior. The creation of this model is an important step towards constructing a continuum model with the intention of running simulations on larger, three-dimensional computational domains. A detailed explanation of the framework on which this model is based can be found in [67–69]. Table 1 defines all variables used in this chapter.

**4.1.1. Lattice and states.** We will first define the discrete biofilm domain using a lattice, where each lattice node can encode the presence of cells and their physiological state. State sets consist of a finite number of elements, each of which represent a condition that the lattice

**Table 1**. **Definitions of the variables used in model.**

| Variable | Definition |
|---|---|
| $\Sigma_B$ | bacterial state set |
| $\Sigma_C$ | stress state set |
| $\mathcal{L}$ | finite lattice that discretizes the domain of interest |
| $s_{\vec{r}}(t)$ | value from $(\Sigma_B, \Sigma_C)$ at lattice node $\vec{r}$ assumed at time t |
| $s_C(\vec{r}, t)$ | value from $\Sigma_C$ at lattice node $\vec{r}$ assumed at time t |
| $s_B(\vec{r}, t)$ | value from $\Sigma_B$ at lattice node $\vec{r}$ assumed at time t |
| $N_L(\vec{r})$ | bacterial interaction neighborhood of lattice node $\vec{r}$ |
| $S\mu_d(\vec{r})$ | stress interaction neighborhood of lattice node $\vec{r}$ |
| $L$ | distance at which non-stressed cells require to divide ($\mu m$) |
| $\mu_d$ | half the width of a cell's interaction neighborhood |
| $a(\vec{r}, t)$ | antibiotic concentration at lattice node $\vec{r}$ at time t |
| $a_1$ | 100X MIC antibiotic concentration |
| $a_2$ | antibiotic flux necessary to fill the domain with |
| | 100X MIC concentration over 1 hour |
| $a_r$ | reduced antibiotic flux from $a_2$ |
| $T_a$ | $A_i$ threshold required for a cell to exhibit the stress response |
| $T_g$ | $A_i$ threshold required to halt a cell's growth |
| $T_d$ | $A_i$ threshold required to fully saturate a cell |
| $dt_a$ | time step for antibiotic diffusion (hour) |
| $dt$ | time step for bacterial growth and stress update (hour) |
| $\gamma_d$ | probability a cell dies in time interval $dt$ |
| $\hat{\gamma}_g$ | maximum probability a cell divides in time interval $dt$ |
| $\gamma_g = \hat{\gamma}_g \cdot f(a, T_a)$ | probability a cell divides in time interval $dt$ |
| $space(N_L(\vec{r}\backslash D))$ | function determines if $\vec{r}$ has adequate space to divide |
| $D$ | doubles $N_L$ size, used for $space(N_L(\vec{r}\backslash D))$ |
| $\alpha(\vec{r})$ | probability of cell placement by an adjacent cell |
| $\beta(\vec{r})$ | probability of stress induction by contact from adjacent cell |
| $\mu_m(\vec{r})$ | probability stress induction via signaling molecules |
| $\Omega$ | computational domain of $\mathcal{L}$ |
| $\partial\Omega$ | boundary of computational domain |

nodes can assume. Let the set $\Sigma_B := \{0, 1\}$ be such that 1 represents the presence of a bacterial cell and 0 represents its absence. Moreover, let the set $\Sigma_C := \{0, 1\}$ be such that 1 represents the expression of a stress response and 0 represents its absence.

**Definition 1** (Biofilm domain). *The discrete biofilm domain is defined by the finite lattice $\mathcal{L} \subset \mathbb{Z}^2$ such that each lattice node $\vec{r} = (r_1, r_2)$ has 4 adjacent nodes. Each node is indexed by the elements of $\mathcal{L}$'s symmetry group and can assume only one value from each set $\Sigma_B$ and $\Sigma_C$ at any given time t.*

We will denote each lattice nodes' state at time $t$ by $s_{\vec{r}}(t) = (s_B(\vec{r}, t), s_C(\vec{r}, t))$ where $s_B$ and $s_C$ are the values assumed by $\vec{r} \in \mathcal{L}$ from $\Sigma_B$ and $\Sigma_C$, respectively. Note that a stress response without the presence of a bacterial cell is not a biologically feasible scenario. Therefore, we exclude the possibility that $s_{\vec{r}}(t) = (0, 1)$. Thus, for any lattice node in the domain, we can construct the dynamical map pictured in Fig 8.

Next, we define the interaction neighborhoods of $\vec{r} \in \mathcal{L}$ for both bacterial cell growth ($N_L$) and stress induction ($S_{\mu_d}$). Interaction neighborhoods are defined as the locations in our domain whose values have influence over the state of a given lattice node.

**Definition 2** (Interaction neighborhoods). *Let the interaction neighborhood of $\vec{r}$ for bacterial cell growth be defined as:*

$$N_L(\vec{r}) = \{\vec{r} \pm \vec{c} \,|\, \vec{c} = (c_1, c_2) \in \mathbb{R}^2, \sqrt{c_1^2 + c_2^2} \le 2L\},$$

*where L is the distance in μm at which less densely packed bacteria grow from each other. Additionally, let the interaction neighborhood of $\vec{r}$ for stress induction be defined as:*

$$S_{\mu_d}(\vec{r}) = \{\vec{r} \pm \vec{c} \,|\, \vec{c} = (c_1, c_2) \in \mathbb{R}^2, c_1 \le \mu_d, c_2 \le \mu_d\} \cap \mathcal{L}$$

*where $\mu_d$ is the maximum distance in μm that stress from a single cell can influence its surroundings.*

The coefficients $\mu_d$ and $L$ are computed using experimental data, by analyzing rigid structure thickness within treated biofilms and is further discussed in Sect 4.4.

**4.1.2. Antibiotics.** Antibiotics are permitted to diffuse along the lattice of our computational domain. Let $a(\vec{r}, t)$ denote the number of antibiotic molecules at location $\vec{r}$ and time $t$. When an antibiotic molecule encounters a lattice node containing a bacterial cell ($s_B(\vec{r}, t) = 1$), the antibiotics become stuck and no longer move through the lattice. This effect models the absorption of antibiotics into the bacterial cell.

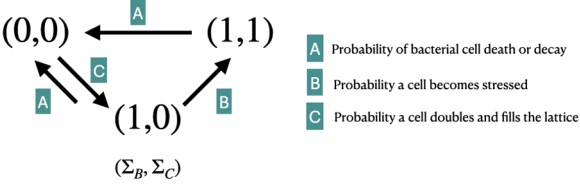

**Fig 8. Map of lattice node dynamics.** Dynamical map of a lattice node. Here, (0,0) indicates the absence of a cell, meaning no stress is applied at that node. (1,0) denotes the presence of an unstressed cell, while (1,1) signifies a stressed cell at the lattice node.

Once a sufficient amount of antibiotic molecules accumulate within a cell and exceeds a threshold $T_a$, the cell exhibits a stress response ($0 \in \Sigma_C \to 1 \in \Sigma_C$). Since erythromycin is a bacteriostatic antibiotic, we defined the threshold $T_g$ to represent the point of antibiotic molecule saturation at which a cell's growth and division is inhibited. Subsequently, when the number of antibiotic molecules reach a third critical threshold, $T_d$, the bacterial cell is fully saturated and any additional antibiotics can freely move past the cell. Note that the stress induction of a bacterial cell whose intracellular antibiotic concentration surpasses the threshold $T_a$ of antibiotic molecules is a deterministic process. In the following section, we will add the stochastic effects of such state transitions.

Supposing that the time scale of reaction is substantially larger than that of diffusion we get the the following partial differential equation:

$$\frac{\partial a(\vec{r}, t)}{\partial t} = D_e \nabla^2 \max(a(\vec{r}, t) - T_d \cdot s_B(\vec{r}, t), 0), \tag{1}$$

where $D_e$ is the effective diffusion coefficient whose computation is shown in Sect 4.4.

In the remaining part of this section let $dt$ denote the time step length of bacterial and stress changes. For further discussion of this time step length and the time scale differences between bacterial, stress, and antibiotic changes, please see Sect 4.4.

**4.1.3. State transitions.** Local update rules define how a given lattice node, $\vec{r}$, transitions from one state in $\Sigma_B \times \Sigma_C$ to another state over a given time interval. Thus, to evolve the system forward in time, we must define local update rules, $R_B : \Sigma_B^{N_L(\vec{r})} \to \Sigma_B$ and $R_C : \Sigma_C^{S\mu_d(\vec{r})} \to \Sigma_C$. Given the information of $s_{\vec{r}}(t)$ and a time step length $dt$, the update rules inform us of the state $s_{\vec{r}}(t + dt)$. For further discussion of any coefficients, see Sect 4.4, where parameter estimations and computations are detailed.

In this section, we will first focus on the update rules pertaining to bacterial growth. An important biological fact, used as the basis of this model, is that these cells are non-motile, and therefore biomass evolution over the simulation occurs via growth, not movement.

**Definition 3** (Bacteria local update rule). *The update rule, $R_B : \Sigma_B^{N_L(\vec{r})} \to \Sigma_B$, can be defined as follows:*

$$R_B(\vec{s}_B(N_L(\vec{r})), t) = \begin{cases} 1, & \text{with probability } W(\vec{s}_B(N_L(\vec{r})) \to 1) \\ 0, & \text{with probability } W(\vec{s}_B(N_L(\vec{r})) \to 0), \end{cases}$$

*where*

$$W(\vec{s}_B(N_L(\vec{r})) \to 0) = \begin{cases} P(1, \in \Sigma_B \to 0 \in \Sigma_B), & \text{if } s_B(\vec{r}, t) = 1, & \text{see eq. (2)} \\ P(0, \in \Sigma_B \to 0 \in \Sigma_B), & \text{if } s_B(\vec{r}, t) = 0, & \text{see eq. (3)} \end{cases}$$

*and*

$$W(\vec{s}_B(N_L(\vec{r})) \to 1) = \begin{cases} P(0 \in \Sigma_B \to 1 \in \Sigma_B), & \text{if } s_B(\vec{r}, t) = 0, & \text{see eq. (4)} \\ P(1 \in \Sigma_B \to 1 \in \Sigma_B), & \text{if } s_B(\vec{r}, t) = 1, & \text{see eq. (5)} \end{cases}$$

We now defined the probabilities that, given one state and its neighborhood, bacteria move to the other. We begin with the process of bacterial death:

$$P(1 \in \Sigma_B \to 0 \in \Sigma_B) = \gamma_d, \tag{2}$$

$$P(1 \in \Sigma_B \rightarrow 1 \in \Sigma_B) = 1 - P(1 \in \Sigma_B \rightarrow 0 \in \Sigma_B) = 1 - \gamma_d, \tag{3}$$

where $\gamma_d$ is the probability a cell will die within the time step $dt$ which, unlike growth, is not dependent on the intracellular antibiotic concentration.

Next, we find the probability that a lattice node receives a bacterial cell from a neighboring node. Cells that are not stressed (i.e., $s_C(\vec{r}, t) = 0$) grow at a distance of $L$ $\mu m$ apart, while cells that are stressed (i.e., $s_C(\vec{r}, t) = 1$) grow at a distance of 1 $\mu m$ apart. Note that this distance is measured from cell center to cell center, implying that growing a distance of 1 $\mu m$ apart is akin to the cells touching. Looking at the neighborhood, $N_L(\vec{r})$, because a cell must grow either 1 or 5 micrometers apart, we find that a bacterial cell can only propagate from 8 positions: those that are adjacent to the cell and those at length $L$ away as illustrated in Fig 9A. This choice of using these positions is consistent with the use of two five-point stencils for numerical discretization: one for the average growth distance under stress and one for the average growth distance not under stress.

Let $\hat{\gamma}_g$ be the maximum probability a cell will double within the time span of $dt$. This quantity can be derived from experimental computations of the doubling time which will be further described in Sect 4.4. Given that erythromycin affects the growth rate of cells, we

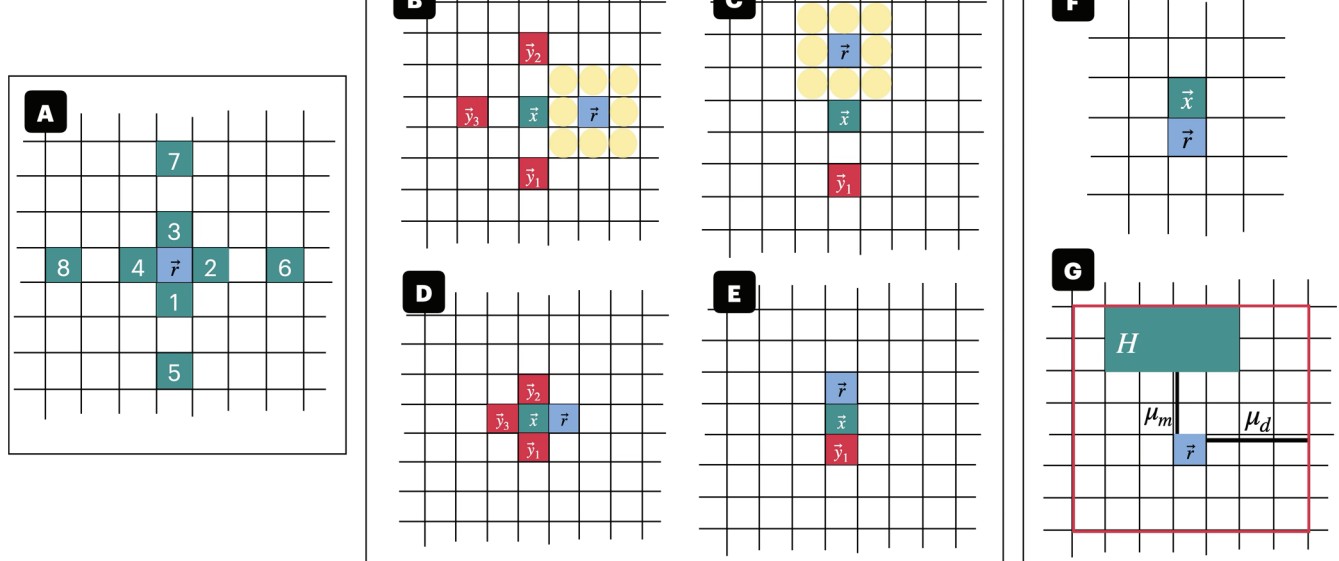

**Fig 9. Bacterial cell placement diagrams. (A)** Assuming the cell to cell distance at which less densely packed bacterial cells require to divide is $L = 3$ $\mu$m. Then the green boxes surrounding lattice node $\vec{r}$ are the locations from which a divided cell placed into $\vec{r}$ can originate from. **(B-E)** Illustration of the probability that a bacterial cell at lattice node $\vec{x}$ divides and places a cell in empty lattice node $\vec{r}$. The yellow boxes denote the lattice nodes space$(N_L(\vec{r})\backslash D, t)$ that ensure the division distance. The lattice nodes denoted by the red squares are the other nodes that are checked for possible bacterial cell occupation. Note that if $\vec{r}$ lies above or below $\vec{x}$ then we do not take into account the lattice nodes to the right or left. If $\vec{r}$ is to the left or right of $\vec{x}$ and if lattice nodes $\vec{y}_2$ or $\vec{y}_1$ are empty then $\vec{x}$ will not place the new cell in $\vec{r}$. $\vec{x}$ will instead place the cell in $\vec{y}_2$ or $\vec{y}_3$ due to the observed growth anisotropy described in Sect 4.1.3. **(B)** $P(\vec{x}$ places cell in $\vec{r}) = \alpha_L^{\text{left}}$. **(C)** $P(\vec{x}$ places cell in $\vec{r}) = \alpha_L^{\text{down}}$. **(D)** $P(\vec{x}$ places cell in $\vec{r}) = \alpha_1^{\text{left}}$. **(E)** $P(\vec{x}$ places cell in $\vec{r}) = \alpha_1^{\text{down}}$. **(F-G)** Illustration of the two stochastic pathways through which stress induction of a cell can take place. **(F)** The blue box, $\vec{r}$, denotes a lattice node such that $s_B(\vec{r}, t) = 1$ and $s_C(\vec{r}, t) = 0$. The green box, $\vec{x}$ denotes a lattice node such that $s_B(\vec{r}, t) = 1$. We write $\beta_{up} = P(\vec{r}$ stressed via $\vec{x}) = \gamma_a \cdot s_C(\vec{x}, t)$. **(G)** Illustration of the quantities necessary to compute the probability that a lattice node, $\vec{r}$, is stressed by signaling molecules. Let the lattice node, $\vec{r}$, be such that $s_C(\vec{r}, t) = 0$ and $s_B(\vec{r}, t) = 1$. Let the green box, $H$, be the set of all lattice nodes containing stressed cells within the neighborhood of $\vec{r}$, i.e. $\forall \vec{x} \in S_{\mu_d}(\vec{r}), \vec{x} \in H$ if $s_C(\vec{x}, t) = 1$. Further, recall $\mu_d$ is the radius of $S_{\mu_d}(\vec{r})$. Then let $\mu_m$ be the minimum distance between $\vec{r}$ and an element of $H$.

calculate the probability that any individual cell will divide by $\gamma_g = \hat{\gamma}_g \cdot f(a(\vec{x}, t), T_g)$ where $f$ is a function determining how the intracellular antibiotic concentration of a cell impacts its growth rate. The function $f$ is a Hill function that is continuously dependent on the antibiotic concentration $a(\vec{x}, t)$ until $a(\vec{x}, t) \geq T_g$ at which point $f \equiv 0$.

Suppose the lattice node of interest is $\vec{r}$ with $s_B(\vec{r}, t) = 0$ and suppose there is a cell located at distance $L$ $\mu m$ in lattice node $\vec{x}$. We then need to assess the probability that a cell propagates from $\vec{x}$ and arrives at lattice node $\vec{r}$. Thus, we ensured there exists sufficient space for a cell in $\vec{x}$ to move into lattice node $\vec{r}$, guaranteeing that no two bacterial cells exist that are less than $L$ $\mu m$ apart. For this, we introduce a space determining function $space(N_L(\vec{r}) \backslash D, t)$.

**Definition 4** (Space determining function). *The function that determines whether there is sufficient space for cells to move from nearby lattice nodes to $\vec{r}$ is defined as*

$$space(N_L(\vec{r}) \backslash D, t) = \begin{cases} 0, & if \; \sum_{i \in N_L(\vec{r}) \backslash D} s_B(i, t) \geq 1, \\ 1, & else, \end{cases}$$

*where*

$$D = \left\{ \vec{r} \pm \vec{c} \mid \vec{c} = (c_1, c_2), L \leq \sqrt{c_1^2 + c_2^2} \leq 2L \right\} \cup \left\{ (0, 0) \right\} \subset N_L(\vec{r}).$$

During biological experiments, we noticed a growth anisotropy. The bacteria within and around the complex structures tends to favor growing up/down. This is a well-documented phenomenon that can be attributed to either optimizing nutrient access [34] or a force of buoyancy [70]. To take this growth anisotropy into account, we say that if the bacterial cell located in $\vec{x}$ has the space available to divide up and down, it will do so. Fig 9B–9E provides illustrations of configurations that $\vec{x}$ and $\vec{r}$ can assume, which have a non-zero probability of cell propagation into $\vec{r}$. Therefore, we must have separate growth probabilities for up/down and left/right growth with a further separation of the division distance. With all this in mind, we can formulate the following separate growth probabilities that a cell in $\vec{x}$ with a corresponding division distance divides and places the new cell into an empty lattice node, $\vec{r}$, which is left/right or up/down from $\vec{x}$:

1. **Left at distance $L$:** $P(\vec{x}$ places cell in $\vec{r}) = \gamma_g \cdot \frac{s_B(\vec{x},t)}{2 - s_B(\vec{y}_3,t)} \cdot s_B(\vec{y}_2, t) \cdot s_B(\vec{y}_1, t) \cdot (1 - s_C(\vec{x}, t)) \cdot$ $space(N_L(\vec{r}) \backslash D, t) = \alpha_L^{\text{left}}$ (Fig 9B). Note, $\alpha_L^{\text{right}}$ is analogous.
2. **Down at distance $L$:** $P(\vec{x}$ places cell in $\vec{r}) = \gamma_g \cdot \frac{s_B(\vec{x},t)}{2 - s_B(\vec{y}_1,t)} \cdot (1 - s_c(\vec{x}, t)) \cdot space(N_L(\vec{r}) \backslash D, t) = \alpha_L^{\text{down}}$ (Fig 9C. Note, $\alpha_L^{\text{up}}$ is analogous.
3. **Left at distance 1:** $P(\vec{x}$ places cell in $\vec{r}) = \gamma_g \cdot \frac{s_B(\vec{x},t)}{2 - s_B(\vec{y}_3,t)} \cdot s_B(\vec{y}_2, t) \cdot s_B(\vec{y}_1, t) \cdot s_C(\vec{x}, t) = \alpha_1^{\text{left}}$ (Fig 9D). Note, $\alpha_1^{\text{right}}$ is analogous.
4. **Down at distance 1:** $P(\vec{x}$ places cell in $\vec{r}) = \gamma_g \cdot \frac{s_B(\vec{x},t)}{2 - s_B(\vec{y}_1,t)} \cdot s_C(\vec{x}, t) = \alpha_1^{\text{down}}$ (Fig 9E). Note, $\alpha_1^{\text{up}}$ is analogous.

Given a lattice node $\vec{r}$, the probability that no new cells move from other lattice nodes to $\vec{r}$ is as follows:

$$\begin{aligned} P(0 \in \Sigma_B \rightarrow 0 \in \Sigma_B) \;=\; & (1 - \alpha_L^{\text{left}}) \cdot (1 - \alpha_L^{\text{right}}) \cdot (1 - \alpha_L^{\text{up}}) \cdot (1 - \alpha_L^{\text{down}}) \cdot \\ & \cdot (1 - \alpha_1^{\text{left}}) \cdot (1 - \alpha_1^{\text{right}}) \cdot (1 - \alpha_1^{\text{up}}) \cdot (1 - \alpha_1^{\text{down}}). \end{aligned} \quad (4)$$

Therefore, the probability that a given lattice node $\vec{r}$ receives a new cell can be defined as:

$$P(0 \in \Sigma_B \to 1 \in \Sigma_B) = P(\vec{r} \text{ receives at least one cell}) = 1 - P(0 \in \Sigma_B \to 0 \in \Sigma_B). \quad (5)$$

Analogously to the local bacteria update rule, we can define the stress update rule, $R_C$, and its corresponding probability statements.

**Definition 5** (Stress local update rule). *The local update rule for stress, $R_C : \Sigma_C^{S_{\mu_d}(\vec{r})} \to \Sigma_C$, can be written as follows:*

$$R_C(\vec{s}_C(S_{\mu_d}(\vec{r})), t) = \begin{cases} 1, & \text{with probability } W(\vec{s}_C(S_{\mu_d}(\vec{r})) \to 1), \\ 0, & \text{with probability } W(\vec{s}_C(S_{\mu_d}(\vec{r})) \to 0), \end{cases}$$

*where*

$$W(\vec{s}_C(S_{\mu_d}(\vec{r})) \to 1) = \begin{cases} P(0 \in \Sigma_C \to 1 \in \Sigma_C), & \text{if } s_C(\vec{r}, t) = 0, \qquad \text{see eq. (8)} \\ P(1 \in \Sigma_C \to 1 \in \Sigma_C), & \text{if } s_C(\vec{r}, t) = 1, \qquad \text{see eq. (11)} \end{cases}$$

*and*

$$W(\vec{s}_C(S_{\mu_d}(\vec{r})) \to 0) = \begin{cases} P(0 \in \Sigma_C \to 0 \in \Sigma_C), & \text{if } s_C(\vec{r}, t) = 0, \qquad \text{see eq. (9)} \\ P(1 \in \Sigma_C \to 0 \in \Sigma_C), & \text{if } s_C(\vec{r}, t) = 1, \qquad \text{see eq. (10)} \end{cases}$$

And as before, we will take a closer look at defining the above probabilities of transition. There exist two types of stress that we encode in this aggregation model: aggregation-mediated cell binding and stress by signaling molecules. Both stress processes ultimately result in a cell shifting from a state $0 \in \Sigma_C \to 1 \in \Sigma_C$ thereby increasing the bacterial cell's local carrying capacity or decreasing their distance required for division.

Cell binding occurs when one stressed cell expressing aggregation substance comes into direct contact with another unstressed cell. The unstressed neighboring cell has a certain probability that it, too, will now express aggregation substance (Fig 9B).

**Definition 6** (Probability of contact stress at single lattice node). *Let $\vec{r}$ be a lattice node such that $s_B(\vec{r}, t) = 1$ and $s_C(\vec{r}, t) = 0$, implying $\vec{r}$ contains an unstressed cell. Suppose there exists a bacterial cell in lattice node $\vec{x} = \vec{r} + (1,0)$ such that $s_B(\vec{x}, t) = 1$. Further, let $\gamma_a$ be the probability of stress induction of an unstressed cell by a single adjacent stressed cell. Then the probability that a cell at $\vec{x}$ causes the stress induction of $\vec{r}$ by contact can be written as*

$$\beta_{up} = P(\vec{r} \text{ stressed via } \vec{x}) = \gamma_a \cdot s_C(\vec{x}, t)$$

*(see Fig 9F). Note that $\beta_{down}, \beta_{left},$ and $\beta_{right}$ are defined analogously.*

Thus we can formulate,

$$P(0 \in \Sigma_C \to 0 \in \Sigma_C \text{ via contact}) = (1 - \beta_{up}) \cdot (1 - \beta_{down}) \cdot (1 - \beta_{left}) \cdot (1 - \beta_{right}), \quad (6)$$

which leads us to

$$P(0 \in \Sigma_C \rightarrow 1 \in \Sigma_C \text{ via contact}) = 1 - P(0 \in \Sigma_C \rightarrow 0 \in \Sigma_C \text{ via contact})$$
$$= 1 - (1 - \beta_{up}) \cdot (1 - \beta_{down}) \cdot (1 - \beta_{left}) \cdot (1 - \beta_{right}). \quad (7)$$

Because the amount of initial rigid structure is constrained by factors other than nutrient availability and continues to increase during the three days of prior biofilm growth, stress by direct contact must occur on the same time scale as the formation of initial complex structures (days). The reconfiguration and rapid increase in the overall bacterial population under antibiotic treatment takes place over the course of 1 hour. Therefore, stress induced by direct contact alone is insufficient to trigger the observed response level. Thus, we defined a rapid form of stress induction. *E. faecalis*, like most bacteria, communicates through various signaling molecules [71,72]. These signaling processes can operate on a time scale consistent with the observed remodeling. Therefore, we consider a second pathway of stress induction, one driven by signaling molecules released by other stressed OG1RF(pCF10) cells. While the exact mechanism of this signaling process remains unclear, it is hypothesized to involve the pCF10 plasmid, chaining, and its associated pheromone signaling molecules [71,72].

It follows from the properties of a classical signaling molecule that the probability of a cell at lattice node $\vec{r}$ becoming stressed by these signaling molecules from already-stressed bacteria should decrease with both the increasing distance from and decreasing density of other stressed cells in its neighborhood $S_{\mu_d}(\vec{r})$. Therefore, we can formulate the probability a cell at lattice node $\vec{r}$ is stressed via signaling molecules.

**Definition 7** (Probability cell at $\vec{r}$ is stressed via signaling molecules). *Let $\mathbf{1}_A\left(S_{\mu_d}(\vec{r})\right)$ be the indicator function that tests the presence of at least one antibiotic-stressed cell within a given neighborhood (i.e. there exists at least one $\vec{x} \in S_{\mu_d}(\vec{r})$ such that $a(\vec{x}) \geq T_a$). We define the minimum distance between the lattice node $\vec{r}$ and a cell expressing aggregation substance to be:*

$$\mu_m(\vec{r}) = \begin{cases} min_{\{\vec{x} \in S_{\mu_d}(\vec{r})|s_C(\vec{x},t)=1\}} \sqrt{(r_1 - x_1)^2 + (r_2 - x_2)^2}, & \text{if } \exists \vec{x} \in S_{\mu_d}(\vec{r}) \text{ s.t. } s_C(\vec{x},t) = 1, \\ \mu_d + 1, & \text{else.} \end{cases}$$

*Recall that with increasing distance from the closest stressed cell to the lattice node $\vec{r}$, we should see a decreasing probability that $\vec{r}$ will undergo stress induction. Therefore we quantify how far $\vec{r}$ is located from a stressed cell:*

$$\psi_1(\vec{r}) = \frac{\mu_d + 1 - \mu_m(\vec{r})}{\mu_d}.$$

*Subsequently, recall that increasing density of stressed cells in the neighborhood of $\vec{r}$ should lead to an increased probability of stress induction for a cell at $\vec{r}$. Thus we quantify the density of stressed cells in the neighborhood of $\vec{r}$:*

$$\psi_2(\vec{r}) = \left( \frac{\sum_{\vec{x} \in S_{\mu_d}(\vec{r})} \left[ s_B(\vec{x},t) \cdot s_C(\vec{x},t) \right]}{|S_{\mu_d}(\vec{r})|} \right).$$

*Let $\gamma_p \leq 1$ be the maximum probability that a cell will undergo stress induction by signaling molecules during the time period dt. Then the probability of a cell at lattice node $\vec{r}$ becoming*

*stressed via signaling molecules is:*

$$P(\vec{r} \text{ stressed via signaling molecules}) = \mathbf{1}_A\left(S_{\mu_d}(\vec{r})\right) \cdot \frac{\gamma_p}{2} \cdot \left(\psi_1(\vec{r}) + \psi_2(\vec{r})\right).$$

This is illustrated in Fig 9G. Note that for simplicity the contribution from the minimum distance to stressed cells and density of stressed cells in the neighborhood of $\vec{r}$ are weighted equally.

It is important to emphasize the role of indicator function $\mathbf{1}_A\left(S_{\mu_d}(\vec{x})\right)$. This function establishes that the signal is in response to the influx of antibiotic molecules while also enabling cells stressed by factors other than their intracellular antibiotic concentration to fortify the signaling molecule process.

Recall that a cell in lattice node $\vec{r}$, that is saturated with antibiotics $(a(\vec{r}, t) \leq T_a))$ becomes stressed with probability 1, which means we only need to define $P(0 \in \Sigma_C \to 1 \in \Sigma_C | a(\vec{r}, t) \leq T_a)$. Under the assumption that stress via signaling molecules and via contact are independent processes we get that

$$P(0 \in \Sigma_C \to 1 \in \Sigma_C | a(\vec{r}, t) \leq T_a) = P(\vec{r} \text{ stressed via signaling molecules})$$
$$+ P(\vec{r} \text{ stressed via contact})$$
$$- \left[P(\vec{r} \text{ stressed via signaling molecules}) \cdot P(\vec{r} \text{ stressed via contact})\right], \quad (8)$$

which leads us to define:

$$P(0 \in \Sigma_C \to 0 \in \Sigma_C | a(\vec{r}, t) \leq T_a) = 1 - P(0 \in \Sigma_C \to 1 \in \Sigma_C | a(\vec{r}, t) \leq T_a). \quad (9)$$

Further, the induction of aggregation substance expression is an irreversible switch. Therefore, for any $\vec{r}$, $1 \in \Sigma_C \to 0 \in \Sigma_C$ if and only if $1 \in \Sigma_B \to 0 \in \Sigma_B$. Thus, for any $\vec{r}$, $1 \in \Sigma_C \to 1 \in \Sigma_C$ if and only if $1 \in \Sigma_B \to 1 \in \Sigma_B$. Ultimately, we get that

$$P(1 \in \Sigma_C \to 0 \in \Sigma_C) = P(1 \in \Sigma_B \to 0 \in \Sigma_B) \quad (10)$$

and

$$P(1 \in \Sigma_C \to 1 \in \Sigma_C) = P(1 \in \Sigma_B \to 1 \in \Sigma_B). \quad (11)$$

It is worth noting that in numerical simulations this probability should only be tested once, meaning if the bacteria is set to die then the stress also becomes 0 at the lattice node. This avoids the biologically infeasible case where a lattice node is labeled as stressed but does not contain a bacterial cell.

**4.1.4. Boundary conditions to model antibiotic entry.** We model the computational domain as a 2D rectangle that represents an x-slice of a 3D microscopy image (see Fig 4). Note that this is the computational version of the previously defined lattice $\mathcal{L}$. The model developed in the previous section is defined on the interior $\Omega$. Boundary conditions on the domain boundaries $\partial\Omega_i$, $i = 1, 2, 3$ are imposed to model antibiotic entry into the domain. We found, after calibrating the model and boundary conditions to experimental imaging data, that $\partial\Omega_2$ and $\partial\Omega_3$ need to be defined as zero-antibiotic flux conditions. From a physical perspective, this condition on $\partial\Omega_2$ captures the robust resistance barrier imposed by eDNA, produced via the formation of initial complex structures, to lie immersed in the extracellular matrix directly above the complex structures. Thus, the presence of an eDNA cloud cover

is only valid above biofilm regions containing complex structures. Therefore, the extent of the computational domain $\Omega$ should align with boundaries of the initial complex structures. Likewise, $\partial\Omega_3$ represents the glass base of an optic bottom well. The details of the specific antibiotic entry conditions modeled by boundary conditions were discussed in Sect 2.

Our model would be incomplete without setting the boundary conditions used for each neighborhood defined in Sect 4.1.1. For the bacteria neighborhood $N_L(\vec{r})$, we implement ghost points outside the domain whose bacterial component is defined as zero. In other words, the model does not allow a cell outside the domain to move into the domain. In contrast, for the neighborhood used to define local stress propagation, we defined $S_{\mu_d}(\vec{r})$ to be $\{\vec{r} \pm \vec{c} | \vec{c} = (c_1, c_2), c_1 \le \mu_d, c_2 \le \mu_d\} \cap \mathcal{L}$, and, therefore, our definition includes what occurs at the boundaries of our domain lattice, $\mathcal{L}$.

With the exception of the antibiotic boundary conditions, which were evaluated and discussed in Sect 2, we have now fully defined our mathematical model. The numerical implementation of this model will be discussed in Sect 4.5.

## 4.2. Biofilm formation, treatment with antibiotics, and imaging

In this study, biofilms were cultured in 24-well 1.5 optical bottom plates (MatTek) under conditions simulating low-sugar environments in the gastrointestinal tract by utilizing 2 $mL$ of Todd-Hewitt (TH) broth without glucose supplementation [73]. *Enterococcus faecalis* strains OG1RF or OG1RF(pCF10) were inoculated from freezer stocks into 1 $mL$ of TH broth and incubated overnight then diluted $1 : 100$ into 1 $mL$ fresh TH in each well. Biofilms were grown at $37°C$ for 3 days, with media replacement twice a day until the samples were ready for imaging. Biofilms grown for 3 days were exposed to 800 $\mu g/mL$ erythromycin (100X MIC) for one hour. Untreated controls were exposed to fresh medium without antibiotic. Prior to imaging, biofilms were washed with 1 mL PBS to remove planktonic cells. The biofilm was stained with 200 $\mu L$ of diluted Syto9 (1 $\mu L$ resuspended in 1 ml PBS) for 10 minutes, followed by another 1 ml PBS wash. Subsequently, 200 $\mu L$ of PBS was added to each well for imaging. Confocal imaging was performed using a Leica SP5 microscope (LCSM) with a 63X objective at a $1024 \times 1024$ resolution to capture 3D biofilm structures. Syto9 fluorescence was excited with a 488 $nm$ laser, and emission was detected in the 495-540 $nm$ range.

## 4.3. Determination of colony forming units (CFU)

Biofilms were grown on glass coverslips in 24-well plates. At day 1, 2, or 3, biofilms were washed 3 times with 1 $mL$ PBS to remove planktonic bacteria. After the third wash, 1 $mL$ of PBS was added to the well. The coverslip and the supernatant (to capture aggregate bacteria dissociated from the coverslip) were placed in a 50 $mL$ conical tube containing 2 $mL$ of PBS (3 $mL$ total volume). The conical tube containing the glass coverslip was kept on ice and sonicated for 30 seconds at setting 7 using the Fisher Scientific 550 Sonic Dismembrator. The sonicated solution (100 $\mu L$) was added to the well containing 900 $\mu L$ of TH media and 8 10-fold serial dilutions were done. Then 10 $\mu L$ of each dilution was spot plated on TH agar. The spot plates were incubated overnight at $37°C$ and counted to determine CFU/biofilm. Counts were determined for 3 replicates each for 3 biological repeats ($n = 9$) at each time point (Day 1, Day 2, and Day 3). Statistical significance was determined by unpaired student's t-test between treated and untreated biofilms.

## 4.4. Parameter estimations and computations

There exist several probabilistic variables whose values we utilized that are dependent on the time step lengths of the numerical simulation. Note, any change in the time step length results in changing all corresponding variables, which was done for the presented computations. Values used for each parameter can be found summarized in Table 2.

**Diffusion coefficient, $D_e$.** The antibiotic diffusion coefficient, $D_e$, was computed to take into account the properties of the porous media of the extracellular matrix. The aqueous diffusion coefficient, $D_{aq}$, is approximately $1 \times 10^{-6} \frac{cm^2}{s}$. We reduced the diffusion coefficient such that $\frac{D_e}{D_{aq}} = 0.2$ [7,75]. Thus, $D_e = 2 \times 10^{-7} \frac{cm^2}{s}$.

**Time interval, $dt_a$.** It is important to first note that the rate of antibiotic diffusion is substantially higher than that of bacterial growth. Therefore, when numerically computing the model, we must use a time splitting scheme. For the numerical simulation of antibiotic diffusion, our corresponding time step is denoted $dt_a$. To simulate the diffusion of antibiotics, we implemented a Forward Euler time-stepping scheme for the time evolution and a central 5-point stencil Finite Difference method for the spatial discretization. We compute the time step $dt_a$ to ensure that the probability for any antibiotic molecule moving to an adjacent lattice node is on average 100% in each time interval:

$$D_e = \frac{\delta^2}{4 \cdot dt_a}$$

where $\delta^2$ is our step size in space and $dt_a$ is our step size in time [76]. Note, $\delta^2 = \delta_y^2 + \delta_z^2$ which are the spatial step sizes in the $y$ and $z$ direction, respectively. Let $\delta_z^2 = 1\mu m^2$ and $\delta_y^2 = 1\mu m^2$ then $\delta^2 = 2\mu m^2$. Further, let $D_e = 2 \times 10^{-7} \frac{cm^2}{s}$. This gives us $dt_a = 0.025s = 7 \times 10^{-6}hr$. This $dt_a$ also corresponds to the maximum time step that satisfies the Courant-Friedrichs-Lewy (CFL) condition of the time-stepping scheme to guarantee numerical stability.

**Maximum probability of cell doubling, $\hat{\gamma}_g$, and cell death, $\gamma_d$.** We conducted a time scale splitting so that bacterial response has a larger time step than that of the antibiotics diffusing, such that $dt = 1000 \cdot dt_a$. Without this scale splitting the probability that a bacterial cell divides would become infinitesimally small at the original time step $dt_a$. Time series assays have shown the doubling time is approximately $0.25hr$, and therefore, in time $dt$, $\hat{\gamma}_g = 0.028$.

**Table 2. Overview of parameters and their values used in numerical simulations.**

| Parameter | Value(s) | Citation/Computation |
|---|---|---|
| $dt_a$ | $7 \times 10^{-6}$ hour | estimated from antibiotic diffusion coefficient through porous media |
| $dt$ | $7 \times 10^{-3}$ hour | estimated from bacterial doubling time |
| $L$ | $5\,\mu m$ | estimated from microscopy data |
| $\mu_d$ | $15\,\mu$m | estimated from microscopy data |
| $T_a$ | 66 antibiotic molecule packages (1 package = 100 molecules) | $1XMIC$ concentration |
| $T_g$ | 1200 antibiotic molecule packages (1 package = 100 molecules) | approximately $20XMIC$ concentration |
| $T_d$ | 10000 antibiotic molecule packages (1 package = 100 molecules) | approximately $150XMIC$ concentration |
| $\gamma_d$ | $\frac{1}{3} \cdot 0.028$ | computed using growth rate and [74] |
| $\hat{\gamma}_g$ | $\frac{4}{3} \cdot 0.028$ | computed from lab data on doubling times |
| $\gamma_a$ | 0.02 | estimated from simulation tuning |
| $\gamma_P$ | 1 | max value taken to favor bacterial survival |

Since our model not only accounts for cell doubling but also cell death this value needs to be adjusted. Doubling times for bacteria tend not to exceed 10 minutes, and the death rate can be assumed to be of same order of magnitude as the growth rate [77]. Accordingly, we estimate $\hat{\gamma}_g = \frac{4}{3} \cdot 0.028$ and $\gamma_d = \frac{1}{3} \cdot 0.028$.

*Maximum probability of cell aggregation, $\gamma_a$.* Cell aggregation by direct contact happens on the order of days. Therefore, to compute $\gamma_a$, we ran numerical simulations to identify values for $\gamma_a$ that would conserve the volume of rigid structures over 1 hour of real-time, in the absence of antibiotic treatment. Given the stochastic nature of our model, this value can be modified slightly without altering the bulk results. The value found was $\gamma_a = 0.02$.

*Average cell growth distance, L.* We estimated the average distance between less densely packed cells using the microscopy data analyzed in Sect 2.2 which lead to $L = 5\mu m$. Several cells that were not chained or attached to a rigid structure were chosen and the distance to their three closest neighbors was measured then averaged. This measurement is conducted with all biofilm microscopy images several times, and the average was found to be approximately $5\mu m$. See Fig 10C for an example of finding the three smallest distances between a cell and its neighbors.

*Maximum cell signaling distance, $\mu_d$.* Using microscopy data, we randomly sampled slices containing rigid structures and measured the width of rigid structures contained in that slice, see Fig 10A. The average of these widths was used to act as a proxy for the diameter of the signaling distance. We found the diameter to be approximately $30\mu m$, and thus the radius, $\mu_d = 15\mu m$.

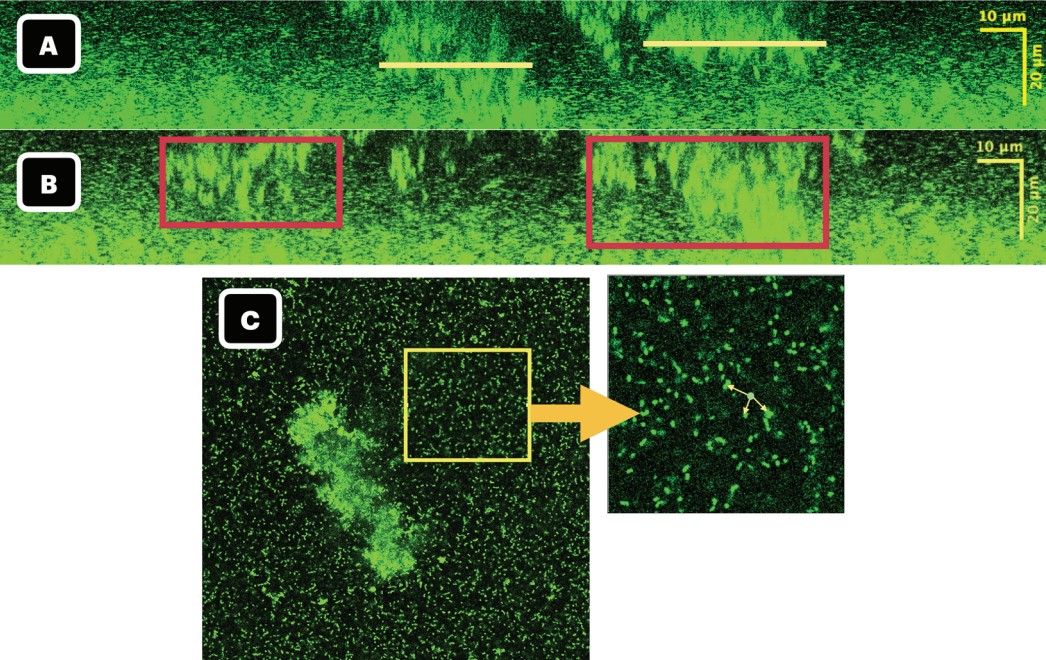

**Fig 10. Measurements of protective regions. (A)** Example of measuring the width ($\mu m$) of rigid structures in a biofilm slice used for quantifying parameter $\mu_d$. Yellow lines register two rigid structures whose lengths are measured and averaged to compute $\mu_d$. The microscopy slice is of a 3-day, 1-hour treated biofilm using method outlined in Sect 4.2.
**(B)** Red boxes encompass protective structures while the area between the two boxes would be the protected region. Under treatment, protective structures buffer and conserve the protected region allowing continued less densely packed, unstressed growth.
**(C)** Example of measuring the distance between four cells that are not chained or part a rigid structure to estimate $L$.

## 4.5. Numerical implementation

The computational domain was discretized into lattice nodes of length $1\mu m$. Since bacterial cells are approximately $1\mu m^3$ in size, each lattice node can have either one cell or no cells inhabiting it. To discretize the model equations on a finite computational grid in space and a finite time grid, we used a central 5-point stencil for the spatial discretization and Forward Euler time-stepping, respectively. Despite the maximum time step for the diffusion operator chosen to satisfy the CFL condition for stability, time splitting remains a critical component of the numerical algorithm, as antibiotics diffuse on a faster time scale compared to bacterial growth. On the interval $dt_a$ probabilities pertaining to bacterial growth and death become arbitrarily small and may be computed as 0. Thus, the time splitting step was not only used to decrease computation time but also to increase computational accuracy.

As mentioned in Sect 4.4 we set the time step of bacterial change $dt = 1000 \cdot dt_a$. For each time-step $dt$ and for each lattice node, the previously detailed probabilities were computed. A random number generator using a uniform distribution is used to evaluate whether or not the probability is reached to change states. The system was updated synchronously except for the case of stress induction. When stress is induced, it is critical to check whether or not the cell will be removed during this iteration, else the simulation will result in lattice nodes taking the value (0,1) which, as previously described, is a non-biological state.

## 4.6. Initial conditions

We used microscopy images of untreated 3-day *E. Faecalis* OG1RF(pCF10) biofilms as our initial conditions. Recall, the goal of this research was to test the sufficiency of initial complex structure formation. Therefore, we needed to use initial conditions which adequately capture a portion of these structures. Since the simulated system is two-dimensional, we constructed an algorithm to evaluate slices of the three-dimensional microscopy stacks. Without loss of generality, we sliced the stacks in the x-direction and called them x-slices. We defined adequate x-slices as those which contain protective structures, thereby offering the opportunity to protect and allow continued growth of the unstressed cells (see Fig 10B). These slices were used to create the initial conditions for the computational simulations.

**4.6.1. Finding adequate x-slices.** While the rigid base layer that forms on $\partial\Omega_3$ must be included in the initial conditions, it needs to be excluded when identifying adequate x-slices. To remove the rigid base layer, tiff images are truncated based on bacterial cell density, thresholded between 25% and 35% of maximum density. Existing Matlab connectivity-based methods were utilized to identify aggregated portions within the complex rigid structures [57]. X-slices containing these aggregated portions were then isolated.

Subsequently, a density-based clustering routine was applied to each x-slice [58]. Our method requires a minimum of 2 clusters, each above a cell threshold $T_c \in \mathbb{N}$. Additionally, the center of mass of each cluster must be sufficiently separated from one another and from $\partial\Omega_1$, with distances defined as thresholds $T_{d_1}$ and $T_{d_2}$, respectively. If these conditions are satisfied the biofilm organization found in the slice has the possibility of surviving treatment.

Remember that for the assumption of eDNA covering to hold true, the z-sliced microscopy data must be appropriately centered. This implies that the aggregated regions of these preformed structures should begin no less than $\mu_d\ \mu m$ away from the nearest side of $\partial\Omega_1$. If the aggregated structures begin closer to the boundary than $\mu_d\ \mu m$, the image cannot be used because the structural changes of such regions would significantly rely on the cellular response outside the modeled domain.

**4.6.2. Adding back less densely packed biofilm.**   To create the simulation initial state, we return to the original microscopy data. Segmenting it along the x-axis, we chose an appropriate x-slice (see Sect 4.6.1). The rigid base layer exclusion is no longer undertaken, and a Matlab connectivity algorithm is employed to detect connected aggregated regions within the complex structures [57]. These aggregated cells are initialized with a stress value of 1, while all remaining image voxels are assigned an initial stress value of 0.

Plasmid-free *E. faecalis* OG1RF biofilms consist of a homogeneous densely packed rigid base overlaid by a very low-density biofilm matrix. This upper layer exhibits notable movement whose biophysical basis is not yet completely understood [78,79]. The microscope used in these studies is a Leica SP5, so the imaging time of a 3D biofilm is about 15-20 minutes. The cell movement in the low density regions of the biofilm is rapid (most commonly 0.06-0.10 $\mu m/s$ resulting in movements of 2-10 $\mu m^3$) [78]. Therefore, the inherent movement in the low density biofilm, coupled with the slow imaging time of the microscope, leads to blurring of the individual cells in the low density biofilm, yielding a low signal intensity. This low signal intensity is thresholded when characterizing aggregated structures, which are accurately quantified at high signal thresholds. It is therefore necessary to reintroduce the less densely packed biofilm uniformly for simulation. The individual cells that can be seen in the lower density biofilm are spaced approximately 5 $\mu m$ apart (Fig 10), so reintroduction involves filling in the less densely packed biofilm at a distance of approximately 5 $\mu m$. The result is an initial condition with defined preformed complex structures and surrounding less densely packed cells.

## Supporting information

**S1 Data. Prism file of Fig 1 data.** This prism file contains the data used to create Fig 1. (ZIP)

**S1 Code. Image processing algorithm.** This Matlab file contains all code used to run the image processing algorithm described in Sect 2.2. (M)

**S2 Data. Fig 2 volume data.** This file contains the volume data plotted in Fig 2. (TXT)

**S3 Data. Fig 2 x-slice data.** This file contains the percentage data plotted in Fig 2. (TXT)

**S2 Code. Fig 5 simulation code.** This Matlab file contains all code used to run the simulations depicted in Fig 5. (M)

**S3 Code. Fig 5 initial condition generation.** This Matlab file contains all code used to constructed the initial conditions depicted in Fig 5. (M)

**S1 Microscopy. Fig 5 x-slice 1.** (TIF)

**S2 Microscopy. Fig 5 x-slice 2.** (TIF)

**S3 Microscopy. Fig 5 x-slice 3.** (TIF)

**S1 Video. Fig 5 video 1.**
(MP4)

**S2 Video. Fig 5 video 2.**
(MP4)

**S4 Data. Fig 6 volume data.** This file contains the volume data plotted in Fig 6.
(TXT)

**S4 Code. Fig 6 simulation code.** This Matlab file contains all code used to constructed the initial conditions depicted in Fig 6.
(M)

**S4 Microscopy. Fig 6 microscopy x-slice 1.**
(TIF)

**S5 Microscopy. Fig 6 microscopy x-slice 2.**
(TIF)

**S6 Microscopy. Fig 6 microscopy x-slice 3.**
(TIF)

**S7 Microscopy. Fig 6 microscopy x-slice 4.**
(TIF)

**S8 Microscopy. Fig 6 microscopy x-slice 5.**
(TIF)

**S9 Microscopy. Fig 6 microscopy x-slice 6.**
(TIF)

**S10 Microscopy. Fig 6 microscopy x-slice 7.**
(TIF)

**S11 Microscopy. Fig 6 microscopy x-slice 8.**
(TIF)

**S12 Microscopy. Fig 2 3D microscopy stack treated 1.**
(TIF)

**S13 Microscopy. Fig 2 3D microscopy stack treated 2.**
(TIF)

**S14 Microscopy. Fig 2 3D microscopy stack treated 3.**
(TIF)

**S15 Microscopy. Fig 2 3D microscopy stack treated 4.**
(TIF)

**S16 Microscopy. Fig 2 3D microscopy stack treated 5.**
(TIF)

**S17 Microscopy. Fig 2 3D microscopy stack treated 6.**
(TIF)

**S18 Microscopy. Fig 2 3D microscopy stack treated 7.**
(TIF)

**S19 Microscopy. Fig 2 3D microscopy stack treated 8.**
(TIF)

**S20 Microscopy. Fig 2 3D microscopy stack untreated 1.**
(TIF)

**S21 Microscopy. Fig 2 3D microscopy stack untreated 2.**
(TIF)

**S22 Microscopy. Fig 2 3D microscopy stack untreated 3.**
(TIF)

**S23 Microscopy. Fig 2 3D microscopy stack untreated 4.**
(TIF)

**S24 Microscopy. Fig 2 3D microscopy stack untreated 5.**
(TIF)

**S25 Microscopy. Fig 2 3D microscopy stack untreated 6.**
(TIF)

**S26 Microscopy. Fig 2 3D microscopy stack untreated 7.**
(TIF)

## Acknowledgments

We would like to thank Isaac Klapper for his invaluable guidance throughout this project. Publication of this article was funded in part by the Temple University Libraries Open Access Publishing Fund.

## Author contributions

**Conceptualization:** Madison Shoraka, Bettina Buttaro, Gillian Queisser.

**Data curation:** Madison Shoraka, Herby Jean-Baptiste, Bettina Buttaro.

**Formal analysis:** Madison Shoraka, Herby Jean-Baptiste, Bettina Buttaro, Gillian Queisser.

**Funding acquisition:** Bettina Buttaro, Gillian Queisser.

**Investigation:** Madison Shoraka, Herby Jean-Baptiste, Bettina Buttaro, Gillian Queisser.

**Methodology:** Madison Shoraka, Bettina Buttaro, Gillian Queisser.

**Project administration:** Bettina Buttaro, Gillian Queisser.

**Resources:** Bettina Buttaro, Gillian Queisser.

**Software:** Madison Shoraka.

**Supervision:** Bettina Buttaro, Gillian Queisser.

**Validation:** Madison Shoraka.

**Visualization:** Madison Shoraka, Herby Jean-Baptiste, Bettina Buttaro.

**Writing – original draft:** Madison Shoraka, Bettina Buttaro, Gillian Queisser.

**Writing – review & editing:** Madison Shoraka, Herby Jean-Baptiste, Bettina Buttaro, Gillian Queisser.

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
