## [Decision Letter · Decision Letter 0]

4 Mar 2025

PCOMPBIOL-D-24-02060

Using stochastic cellular automata to model and define sufficient conditions for the survival of Enterococcus faecalis  biofilms with the pCF10 plasmid under erythromycin treatment

PLOS Computational Biology

Dear Dr. Shoraka,

Thank you for submitting your manuscript to PLOS Computational Biology. After careful consideration, we feel that it has merit but does not fully meet PLOS Computational Biology's publication criteria as it currently stands. Therefore, we invite you to submit a revised version of the manuscript that addresses the points raised during the review process.

Please submit your revised manuscript within 60 days May 04 2025 11:59PM. If you will need more time than this to complete your revisions, please reply to this message or contact the journal office at ploscompbiol@plos.org. Please include the following items when submitting your revised manuscript:

We look forward to receiving your revised manuscript.

Kind regards,

Ricardo Martinez-Garcia

Academic Editor

PLOS Computational Biology

Amber Smith

Section Editor

PLOS Computational Biology

**Journal Requirements:**

1) Please ensure that the CRediT author contributions listed for every co-author are completed accurately and in full.At this stage, the following Authors/Authors require contributions: Madison Shoraka, Herby Jean-Baptiste, Bettina Buttaro, and Gillian Queisser. Please ensure that the full contributions of each author are acknowledged in the "Add/Edit/Remove Authors" section of our submission form.The list of CRediT author contributions may be found here: https://journals.plos.org/ploscompbiol/s/authorship#loc-author-contributions 2) We ask that a manuscript source file is provided at Revision. Please upload your manuscript file as a .doc, .docx, .rtf or .tex. If you are providing a .tex file, please upload it under the item type u2018LaTeX Source Fileu2019 and leave your .pdf version as the item type u2018Manuscriptu2019. 3) We have noticed that you have uploaded Supporting Information files, but you have not included a list of legends. Please add a full list of legends for your Supporting Information files after the references list. 4) Thank you for stating that  "Data and associated code is located on Github found at: https://github.com/Madison-Shoraka/E_faecalis_erm_treatment.git. This information will be publicly available upon publication." Please note that, though access restrictions are acceptable now, your entire minimal dataset will need to be made freely accessible if your manuscript is accepted for publication. This policy applies to all data except where public deposition would breach compliance with the protocol approved by your research ethics board. If you are unable to adhere to our open data policy, please kindly revise your statement to explain your reasoning and we will seek the editor's input on an exemption. 5) Please amend your detailed Financial Disclosure statement. This is published with the article. It must therefore be completed in full sentences and contain the exact wording you wish to be published.1) State the initials, alongside each funding source, of each author to receive each grant. For example: "This work was supported by the National Institutes of Health (####### to AM; ###### to CJ) and the National Science Foundation (###### to AM)."2) State what role the funders took in the study. If the funders had no role in your study, please state: "The funders had no role in study design, data collection and analysis, decision to publish, or preparation of the manuscript." 6)  [Fig8_1_flux_reduction_video_supplemental1.mp4 and Fig8_2_flux_reduction_video_supplemental2.mp4 ] are currently uploaded as file type “Other”, which is not viewable by the reviewers. Please change the file types to 'Supporting Information' and include a legend in the manuscript if you wish them to be included in review. 

**Reviewers' comments:**

Reviewer's Responses to Questions

**Comments to the Authors:**

**Please note that one of the reviews is uploaded as an attachment.**

Reviewer #1: The manuscript "Using stochastic cellular automate to model and define sufficienct conditions for the survival of Enterococcus faecalis biofilms with the pCF10 plasmid under erythromycin treatment" is really thoroughly and well written.

The problem is clearly stated, all procedures are well-described and reasonable.

The model approach with the CA for the bacteria (and stress level) combined with a diffusion for the antibiotics is not absolutely innovative per se, but is set up also thoroughly considering all necessary details, and serving well for the purposed aimed in here.

There are only minor points which should be considered for a final version:

* There is some (partially word-by-word) identity between abstract and introduction, would be great to have it not exactly word-by-word the same but adapted to the purpose, respectively.

* As the model is in principle a hybrid setting (and not "only a CA"), one could mention that also in the title or at least in the abstract or so.

* Is it correct (following Fig. 1) - it is not possible in the setting to loose stress again?

* In the context of subsection 2.1.2: Have the antibiotics been represented on the same grid as the CA itself? Which means, the main difference between the antibiotics versus cells and stress level is that the first one can assume continuous values, whereas the second ones have the discrete states with 0 and 1? Has it been checked if that setting (including the time step size) would be ok concerning stability criteria for finite differences for the antibiotics?

(Maybe I have overlooked the concrete equations for the antibiotics, but if they are indeed not stated explicitly, this would be good to include, maybe in an appendix if it is too detailed for the main part of the paper)

* the within-figure-numbering in Fig. 2 is a bit complicated, with have two different subfigures A and B etc.

* in caption of Fig. 3, I think the "x-slice" should be typed with a small x, too, as directly above and around.

* at the beginning of the results part, section 3.1 (lines 429 ff), this is again doubling partially the abstract. I think, would be good, which basics to introduce where, but once is enough (or at least not more or less identical)

Reviewer #2: review is uploaded as an attachment.

Reviewer #3: In the manuscript 'Using stochastic cellular automata to model and define sufficient conditions for the survival of Enterococcus faecalis biofilms with the pCF10 plasmid under erythromycin treatment,' the authors present a computational and experimental study investigating how biofilm structure influences the survival of E. faecalis under antibiotic stress. They develop a stochastic cellular automata model that incorporates antibiotic diffusion, bacterial growth dynamics, and spatial constraints to simulate biofilm responses to erythromycin. The study combines microscopy-based observations with simulations to argue that biofilm survival depends on both preformed structural complexity and an eDNA cloud that shields bacteria from antibiotic penetration. While the manuscript introduces an interesting modeling framework and provides insights into biofilm resilience, it presents several major issues that require clarification before publication, including problems in the experimental methods and modeling assumptions that need further justification.

Major concerns:

* The authors convincingly demonstrate that initial biofilm structure alone is not sufficient for survival. However, the argument that an eDNA cloud is sufficient seems more speculative. While eDNA clearly plays an important role, it is unclear whether it alone can ensure survival. Have the authors tested conditions where biofilms with eDNA but without complex preformed structures exhibit similar survival? If not, it may be more accurate to state that eDNA is necessary but not necessarily sufficient.

* The definition and role of the stress variable in the model are unclear. While bacterial cells and antibiotic molecules are well-defined, 'stress' seems more abstract. What biological process does this variable represent? Does it correspond to a measurable quantity, and if so, what are its units? Additionally, the way stress influences bacterial growth and division appears to overlap with the direct effects of antibiotic exposure—would it be clearer to model growth inhibition as a function of intracellular antibiotic concentration instead?

Modeling assumptions:

* I am unsure why the authors model erythromycin action as a threshold-based effect rather than a continuous dose-dependent response (line 122). Given that erythromycin is bacteriostatic, one would expect a gradual inhibition of protein synthesis rather than a strict transition from unstressed to stressed at a defined antibiotic concentration. Can the authors clarify the biological reasoning behind this assumption? Would a graded response better capture the heterogeneity of bacterial stress responses under antibiotic pressure?

* The assumption that erythromycin halts both growth and death seems biologically unrealistic. While erythromycin is bacteriostatic and inhibits growth, cell death can still occur due to factors such as metabolic stress, prolonged inhibition, or cell lysis. The distinction between 'death' and 'decay' is unclear—what exactly does decay represent in the model? If decay is functionally equivalent to death, does the model account for varying survival rates within biofilm subpopulations? A more biologically realistic approach might include a low but nonzero death rate that accounts for metabolic failure or prolonged antibiotic exposure.

* As far as I can tell, the model is implemented in two dimensions. Given that biofilms are inherently three-dimensional, can the authors comment on how this limitation might affect their results? For example, does the model adequately capture antibiotic diffusion and eDNA-mediated protection in a way that would still hold in 3D? If so, what justifies this assumption? If not, what key insights might change in a full 3D model?

* The model assumes that stressed cells grow at a distance of 1 µm apart, implying a reduction in cell size during treatment. However, I could not find experimental data supporting this assumption—does erythromycin exposure actually cause a physical reduction in cell size? If this is based on empirical observation, could the authors provide quantitative evidence (e.g., microscopy measurements)? If this is purely a modeling assumption, how do they justify it biologically?

Experimental model system:

* The claim that CFUs double within an hour under erythromycin treatment is quite strong, especially given that erythromycin is bacteriostatic. The authors state that this increase is due to growth rather than movement or dispersal, but I am not sure the evidence fully supports this. Could they provide additional experimental validation to rule out alternative explanations such as biofilm dispersal, changes in culturability, or sampling variation? If direct evidence of growth (e.g., time-lapse imaging, biomass quantification, or DNA synthesis assays) is available, it would strengthen this claim considerably.

* The manuscript does not clearly define what 'untreated' means in the experimental setup. Were these biofilms grown in parallel under identical conditions, with the only difference being the absence of erythromycin? Were they sampled at the same time points? This is important for interpretation—if untreated biofilms also exhibit CFU increases, then the role of erythromycin in driving biofilm remodeling may be less clear. Can the authors clarify how the untreated control was designed and justify that it is a valid comparison?

* The existence of an eDNA cloud appears to be supported primarily by the microscopy image in Figure 7. However, it is unclear whether this is sufficient evidence to confirm both the presence and functional role of eDNA in biofilm survival. Could the authors provide additional validation, such as quantification of extracellular DNA, DNase treatment experiments, or staining controls in untreated biofilms? Without these, it is difficult to determine whether the observed fluorescence represents functional eDNA rather than an artifact of cell lysis or antibiotic exposure.

* The use of 100× MIC erythromycin seems quite high, and I could not find a clear justification for this concentration in the manuscript. How was this critical dose determined? Since biofilms are known to have diffusion barriers and antibiotic resistance mechanisms, what would be the expected erythromycin concentration inside the biofilm under these conditions? Have the authors considered performing a dose-response experiment to determine whether their findings are robust across a range of more clinically relevant antibiotic concentrations?

Minor comments:

* The manuscript would benefit from more consistent strain nomenclature. The first mention should use the full genus and species name (e.g., Enterococcus faecalis OG1RF), and subsequent mentions should abbreviate the genus (E. faecalis OG1RF). Additionally, bacterial names should always be italicized. Standardizing this throughout the manuscript will improve clarity and readability.

* The manuscript uses 'demonstrate' for both experimental (Line 29) and computational (Line 58) results. Since 'demonstrate' implies definitive proof, especially in mathematics, it may not be the best choice for computational findings.

* The manuscript provides a GitHub link for the code and supplementary videos, but the repository does not appear to contain any public files. Since the code is essential for assessing the computational model and reproducibility, could the authors ensure that the files are accessible to reviewers? If the repository is private, an alternative way to access the files would be helpful.

**Have the authors made all data and (if applicable) computational code underlying the findings in their manuscript fully available?**

Reviewer #1: **No: *** There is a github-link mentioned on line 631, but this does not seem to work yet, so I cannot check, if everything is available, but would guess it will be then (?) . So my comment about this can probably be ignored

Reviewer #2: **No: **I was not able to find the raw confocal microscopy images for the experimental results or a reference to the data being deposited anywhere. The github with the code is not available yet, but this is of course understandable. Data provided for individual figures in supplement is lacking a read me or some explanatory text about what it is.

Reviewer #3: **No: **The GitHub repository provided by the authors does not contain publicly accessible files, preventing evaluation of the computational model. Additionally, the manuscript does not specify whether all raw data, including microscopy images and CFU counts, are available.

PLOS authors have the option to publish the peer review history of their article (what does this mean?). If published, this will include your full peer review and any attached files.

Reviewer #1: No

Reviewer #2: No

Reviewer #3: No

**Figure resubmission:**
---

## [Decision Letter · Decision Letter 1]

22 Jul 2025

PCOMPBIOL-D-24-02060R1

Using stochastic cellular automata to model and define sufficient conditions for the survival of Enterococcus faecalis  biofilms with the pCF10 plasmid under erythromycin treatment

PLOS Computational Biology

Dear Dr. Shoraka,

Thank you for submitting your manuscript to PLOS Computational Biology. After careful consideration, we feel that it has merit but does not fully meet PLOS Computational Biology's publication criteria as it currently stands. Therefore, we invite you to submit a revised version of the manuscript that addresses the points raised during the review process.

Please submit your revised manuscript within 30 days Sep 21 2025 11:59PM. If you will need more time than this to complete your revisions, please reply to this message or contact the journal office at ploscompbiol@plos.org. Please include the following items when submitting your revised manuscript:

We look forward to receiving your revised manuscript.

Kind regards,

Ricardo Martinez-Garcia

Academic Editor

PLOS Computational Biology

Amber Smith

Section Editor

PLOS Computational Biology

**Journal Requirements:**

1) Your manuscript's sections are not in the correct order.  Please amend to the following order: Abstract, Introduction, Results, Discussion, and Methods

**Reviewers' comments:**

Reviewer's Responses to Questions

Reviewer #1: From my point of view, the authors have addressed the points raised up in the first version/review adquately.

I could imagine some details to be done differently, but this is more a matter of taste.

A very little comment:

In Definition 7, it looks like something went wrong with the Math mode, could be corrected (no further review necessary for that).

Reviewer #2: Overall, most of my questions were appropriately addressed in the manuscript. The restructuring of the results also helps in broadening the impact of the paper to a wider audience.

Some considerations that should be fixed.

- In line 111, there is a reference to Fig 1G-R, but the figure only has G panels.

- In section 2.3 a brief introduction of the cellular automaton model should be provided, to guide to reader in understanding the boundary conditions, and subsequent results. Ideally before line 193, as there is reference to sufficiency conditions in the model without having introduced the model. Which makes the paper read out of order.

- In line 277, this statement is very strong. Would recommend to switch to: Taken together our results strongly suggest that initial complex structures …

- In regards to line 283, and throughout the document, the word demonstrated usually implies mathematical derivation, or proven results in the literature, therefore when referring to experimental results it might be more appropriate to write experimentally observed. E.g. “capture the mechanisms pertaining the experimentally observed rearrangement of …”

- Include the variable a (100x MIC concentration) in one of the parameter tables.

**Have the authors made all data and (if applicable) computational code underlying the findings in their manuscript fully available?**

Reviewer #1: **No: **In the response to the previous review, they stated that they would make it public for the publication. If they really do so, not optimal, but ok from my point of view.

Reviewer #2: Yes

PLOS authors have the option to publish the peer review history of their article (what does this mean?). If published, this will include your full peer review and any attached files.

Reviewer #1: No

Reviewer #2: No

**Figure resubmission:**
---

## [Editor Report · Decision Letter 2]

11 Aug 2025

Dear Shoraka,

We are pleased to inform you that your manuscript 'Using stochastic cellular automata to model and define sufficient conditions for the survival of Enterococcus faecalis  biofilms with the pCF10 plasmid under erythromycin treatment' has been provisionally accepted for publication in PLOS Computational Biology.

Best regards,

Ricardo Martinez-Garcia

Academic Editor

PLOS Computational Biology

Amber Smith

Section Editor

PLOS Computational Biology

---

## [Editor Report · Acceptance letter]

PCOMPBIOL-D-24-02060R2

Using stochastic cellular automata to model and define sufficient conditions for the survival of Enterococcus faecalis  biofilms with the pCF10 plasmid under erythromycin treatment

Dear Dr Shoraka,

I am pleased to inform you that your manuscript has been formally accepted for publication in PLOS Computational Biology. Your manuscript is now with our production department and you will be notified of the publication date in due course.

With kind regards,

Lilla Horvath
